# EVOLvE: Evaluating and Optimizing LLMs For In-Context Exploration

Allen Nie [* 1]  Yi Su [* 2]  Bo Chang [* 2]  Jonathan N. Lee [2]  Ed H. Chi [2]  Quoc V. Le [2]  Minmin Chen [2]

## Abstract

Despite their success in many domains, large language models (LLMs) remain under-studied in scenarios requiring optimal decision-making under uncertainty. This is crucial as many real-world applications, ranging from personalized recommendations to healthcare interventions, demand that LLMs not only predict but also actively learn to make optimal decisions through *exploration*. In this work, we measure LLMs' (in)ability to make optimal decisions in bandits, a state-less reinforcement learning setting relevant to many applications. We develop a comprehensive suite of environments, including both context-free and contextual bandits with varying task difficulties, to benchmark LLMs' performance. Motivated by the existence of optimal exploration algorithms, we propose efficient ways to integrate this algorithmic knowledge into LLMs: by providing explicit *algorithm-guided support* during inference; and through *algorithm distillation* via in-context demonstrations and fine-tuning, using synthetic data generated from these algorithms. Impressively, these techniques allow us to achieve superior exploration performance with smaller models, surpassing larger models on various tasks. We conducted an extensive ablation study to shed light on various factors, such as task difficulty and data representation, that influence the efficiency of LLM exploration. Additionally, we conduct a rigorous analysis of the LLM's exploration efficiency using the concept of regret, linking its ability to explore to the model size and underlying algorithm.

## 1. Introduction

The rapid advance of LLMs has positioned them as valuable tools for a wide range of decision-making tasks, including but not limited to personal assistants (Liu et al., 2024a), recommendation systems (Li et al., 2023a), game-playing (Wang et al., 2023a;c), education (Nie et al., 2024; He-Yueya et al., 2024), and healthcare (Singhal et al., 2023). In these tasks, LLMs function as agents that engage with users or the environment in a dynamic interaction process. For example, at each time step, the LLM suggests a pedagogical strategy or make a recommendation to a specific user, then receives feedback - either explicit or implicit - in the form of rewards. Based on this feedback, the agent updates its beliefs about the environment, e.g., underlying reward distributions, and adapts its strategies to maximize the cumulative reward. These tasks differ fundamentally from classic prediction tasks where LLM is used to predict a target. A decision making LLM only receives partial feedback, i.e., the reward for its own actions, but not for others. Thus, it requires the LLM to effectively interact with the environment and *explore* to discover the optimal action. Meanwhile, exploring an unknown action that turns out to have lower reward than the known ones incurs an opportunity cost. The agent, therefore, needs to strike a balance between exploration and exploitation. While the exploration-exploitation tradeoff has been extensively studied in the pre-LLM era, particularly in the fields of bandits (Li et al., 2010; Slivkins et al., 2019) and reinforcement learning (Mnih, 2013; Osband et al., 2013; Sutton, 2018), it remains unclear how LLMs approach this tradeoff when faced with uncertainty.

We study LLMs' *in-context exploration* capabilities under the simplified framework of bandits — a stateless form of reinforcement learning that is highly applicable to many domains. We set up the LLM to interact with the environment over $T$ rounds. In each round, it receives the full history of its past interactions, the current state (if provided), and a set of actions, and it is tasked with selecting an action to maximize the cumulative reward. Ideally, the LLM should adaptively choose an action in each round to learn the reward distributions of different actions and eventually converge to consistently selecting the optimal one. We study LLM's ability to do so *in-context*, without the need to re-train, which we dubbed as *in-context exploration*. Unlike using LLM for reward-free or curiosity-driven exploration, *in-context exploration* is in-context self-improvement, where the LLM builds up context through interactions to solve a task.

---
[*]Equal contribution [1]Stanford University [2]Google DeepMind. Correspondence to: Allen Nie <anie@cs.stanford.edu>, Yi Su <yisumtv@google.com>.

*Proceedings of the 42nd International Conference on Machine Learning*, Vancouver, Canada. PMLR 267, 2025. Copyright 2025 by the author(s).

We introduce *BanditBench*[1], a comprehensive suite of multi-armed bandit (MAB) (Slivkins et al., 2019) and contextual bandit (CB) (Li et al., 2010) environments *in natural language* to rigorously evaluate the decision-making capabilities of LLMs. Building on the pioneering work of Krishnamurthy et al. (2024), we significantly expand the benchmark by incorporating a broader range of tasks with varying complexities, including variations in the number of arms, reward distributions, exploration difficulty, and textual descriptions of environments. Additionally, we extend it to CB environments, where rewards across arms depend on contextual features, to assess generalization in LLM exploration.

To enhance LLMs for in-context exploration, we leverage known bandit algorithms such as Upper-Confidence Bound (UCB) algorithm, which have been proven "optimal" under mild conditions. We investigate two approaches: (1) *inference-time algorithm-guided support*, where summary statistics on interaction history, along with descriptions of bandit algorithms, are provided in context for LLMs to choose actions, and (2) *algorithm distillation via optimal demonstration data*, where "oracle" trajectories from optimal bandit algorithms are provided as either *in-context few-shot demonstrations* or *optimal behavior fine-tuning*. We benchmark off-the-shelf LLMs of different sizes - both open-sourced and proprietary - and those enhanced by our approaches on *BanditBench*. Both approaches demonstrate promising improvements over baseline methods that rely solely on raw interaction histories presented as sequences of (action, reward) tuples. Furthermore, our results show that fine-tuning to distill optimal exploration behavior leads to strong generalization across domains, enabling smaller models to achieve superior exploration performance compared to larger models. We also perform extensive ablation studies that reveal how training task difficulty, textual representation and Algorithm-Guided Support impact model performance. To gain deeper insights into the exploration efficiency of different methods, we fit a parametric function to the observed regret patterns, allowing for a more rigorous interpretation of the exploration efficiencies of various LLMs and our proposed approaches.

## 2. Related Work

Several prior works have investigated the use of LLMs for decision-making. In one category, numerous studies have deployed LLMs directly as agents in decision-making problems such as games (Yao et al., 2023; Brooks et al., 2024; Shinn et al., 2024; Wang et al., 2023a; Xi et al., 2023). However, fewer works have systematically evaluated LLMs' capabilities in general decision-making setup, especially in

relation to classical concepts in decision-making like exploration. Our work extends the research of Krishnamurthy et al. (2024), who examined LLMs' exploration capabilities in small-scale MAB tasks. Their findings, which showed positive results only with substantial intervention, are consistent with our broader analysis across both MAB and CB tasks at various scales. Mirchandani et al. (2023); Rahn et al. (2024); Felicioni et al. (2024) also evaluated the ability of LLMs to learn in-context and solve bandit-like decision-making problems.

Another relevant line of research focuses on in-context learning for decision-making and reinforcement learning (RL) with domain-specific transformers. Laskin et al. (2022) distilled demonstrations from RL algorithms into a transformer and showed that it learns to imitate the RL process to solve new RL tasks. Similarly, Lee et al. (2024) trained transformers with optimal action labels, showing that the model learns to execute posterior sampling for RL (Osband et al., 2013) in-context. This area has been further studied by Raparthy et al. (2023); Lin et al. (2023); Bai et al. (2023). However, these studies focus on domain-specific decision-making, whereas our paper examines general-purpose decision-making capabilities in language models. Our inference-time algorithm-guided support shares a similar conceptual framework with recent efforts to align LLMs at inference time. These include employing explicit value functions as prefix scorers (Mudgal et al., 2023), and leveraging both implicit and explicit value functions to guide decoding (Liu et al., 2024b). In the realm of algorithm distillation, much of the research on LLMs has concentrated on chain-of-thought (CoT) reasoning (Wang et al., 2023b; Li et al., 2023b), while (Gandhi et al., 2024) focused on search and backtracking. Our work focuses on distilling a more complex class of algorithms that involve uncertainty estimation and linear regression.

The motivation of our work is also inspired by the cognitive science literature on human. Prior studies have found that humans balance exploration and exploitation through a mix of directed and random exploration (Wilson et al., 2014), and that these behaviors are supported by distinct neural mechanisms (Daw et al., 2006). Gershman (2018) further decomposes human exploration into algorithmic components. These works provide future directions for evaluating LLM's behavior in addition to reward (Pan et al., 2025). Additionally, efficient exploration also benefits other areas of RL, such as multi-agent collboration (Qu et al., 2024) and preference optimization (Bai et al., 2025).

## 3. In-Context Exploration

In this section, we define the problem of In-Context Exploration (ICE), following the setup in (Krishnamurthy et al., 2024). An agent interacts with an environment by observing

---

[1]Github: https://github.com/allenanie/EVOLvE. You can install the code with: pip install banditbench

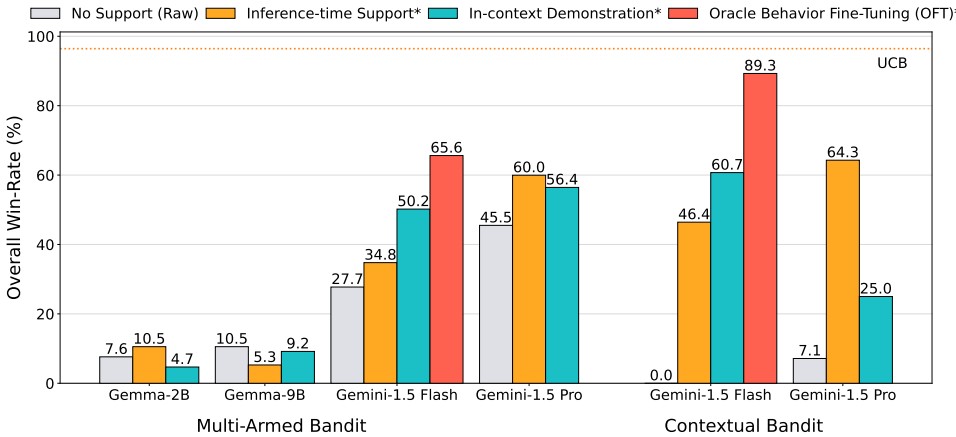

Figure 1: The best achieved performance within each category of method in both MAB and CB. Note that we took the max over different methodology setups within each category. For contextual bandit, OFT enabled a small fine-tuned model (Gemini-1.5 Flash) to approach the performance of the optimal classical algorithm (UCB).

state information, selecting actions, and collecting feedback. The goal of the agent is to maximize its cumulative reward through multiple rounds of interactions. Specifically for ICE, the agent is an LLM that keeps a history of observations and interactions with the environment in its context. The agent determines its actions based on this context, rather than by updating its weights or executing hand-designed exploration strategies.

**Notation and Definitions.** We primarily consider *bandits*, a simple class of environments that still incorporates many fundamental challenges in decision-making. Here, we describe a framework that encompasses both *multi-armed bandits (MAB)* and *contextual bandits (CB)*. A bandit environment $\mathcal{T}$ is defined as $\mathcal{T} = (\mathcal{X}, \mathcal{A}, R)$, where $\mathcal{A}$ defines a set of valid actions. $\mathcal{X}$ is the set of state information (if any), and $R$ represents the underlying reward distributions of actions, which are unknown to the agent. MAB and CB tasks differ in whether the context $x$ is provided and used: in MAB, the reward depends solely on the action, whereas in CB it depends on both the action and the context. The interaction between the agent and the environment occurs over $T \in \mathbb{N}$ steps. At each time step $t \in [T]$, the environment reveals a new observation[2] $x_t \in \mathcal{X}$, the agent selects an action $a_t \in \mathcal{A}$ following its policy $\pi$, and then a reward $r_t^{a_t} \sim R^{a_t}(x_t)$ is revealed. Given an LLM agent with policy $\pi$, it determines its action $a_t \sim \pi(H_t^{\pi})$, where $H_t^{\pi} = (x_1, a_1, r_1^{a_1}, \ldots, x_t)$ stores the historical actions taken by the agent and the corresponding environment feedback, which is sent as input context to the LLM.

Over $T$ rounds, we measure the performance of an agent

$\pi$ on task $\mathcal{T}$ as its expected cumulative reward, given by $J_{\mathcal{T}}(\pi) = \mathbb{E}_{\mathcal{T}, \pi} \left[ \sum_{t=1}^{T} r_t^{a_t} \right]$. The optimal policy $\pi^*$ represents the agent that selects the action with the highest average reward, defined as $\pi^*(x) = \arg\max_a \mathbb{E}_{\mathcal{T}} [r^a \mid x]$. A commonly used metric to measure the performance of an agent or algorithm is regret.

**Definition 1** (Cumulative Regret). *The expected regret of a policy $\pi$ under task $\mathcal{T}$ is: $REG(\pi) = \mathbb{E}_{\mathcal{T}, \pi} \left[ \sum_{t=1}^{T} (r_t^{a_t^*} - r_t^{a_t}) \right] = J_{\mathcal{T}}(\pi^*) - J_{\mathcal{T}}(\pi)$, where $a_t^* = \pi^*(x_t)$.*

We expect *good* agents to have *average* regret that converges to zero (i.e. $\frac{1}{T}REG \xrightarrow{T} 0$), demonstrating that they eventually learn to perform as good as the optimal policy. UCB and Thompson Sampling are two such examples with sublinear regret. Examples of cumulative regret curves are shown in Figure A4c.

**Representing Histories in Context.** Developing an LLM agent suited for in-context decision-making tasks also requires designing a robust contextualization function $\phi$ that translates histories $H_t^{\pi}$ for the LLM to consume. The obvious baseline for $\phi$ is to simply record the **Raw History** (**RH**) from the environments as a list of (context, action, reward) tuples directly as the context. In this representation, the context length of $\phi(H_t^{\pi})$ grows linearly with $t$, and **RH** contains all information. While **RH** is a general contextualization function applicable to any task $\mathcal{T}$, more advanced task-specific contextualization functions may exist and yield better performance. For example, (Krishnamurthy et al., 2024) proposed a **Summarized History** function (**SH**) that compresses the history while still containing sufficient information for a given task $\mathcal{T}$. **RH** and **SH** differ in how past interaction histories are represented to the LLM agent, as

---

[2]In CB, context $x$ is exogenous and independently sampled from a stationary distribution; it is not affected by action $a$, as in the full RL setting.

shown in Figure A2. At time step $t$, **RH** provides a complete list of past interactions as (Time $t'$, Action Name $a_{t'}$, Reward $r_{t'}$) for $t' = 0 \cdots t$. In contrast, **SH** provides sufficient statistics of the past interactions. Specifically, under MAB, **SH** utilizes the empirical mean over each arm (i.e., $\hat{\mathbb{E}}[r^a], \forall a \in \mathcal{A}$), the number of times each arm has been pulled up to time $t$, $N_t(a)$, and the current horizon $t$. In this paper, we consider good contextualization to be those that satisfy "sufficiency", which we define as:

**Definition 2** (Sufficient Contextualization). *Given a policy class $\Pi$, let $\Pi^\phi \subset \Pi$ and $\Pi^{raw} \subset \Pi$ be the sets of policies that take a history representation $\phi(H_t)$ using the contextualization function $\phi$ and the raw history $H_t$, respectively. Then the contextualization function $\phi$ is sufficient if*

$$\lim_{T \to \infty} \left[ \inf_{\pi^\phi \in \Pi^\phi} \frac{1}{T} REG(\pi^\phi) - \inf_{\pi^{raw} \in \Pi^{raw}} \frac{1}{T} REG(\pi^{raw}) \right] = 0.$$

In other words, the best agent that uses the history representation can asymptotically achieve the same average regret as one with the full raw history, meaning that the contextualization preserves all the essential information needed for effective decision-making.

## 4. BanditBench

We present BanditBench, an extensive suite of MAB (Slivkins et al., 2019) and CB (Li et al., 2010) environments in *natural language* to benchmark the in-context exploration capabilities of LLMs. We show two examples in Figure A1. A wide range of real-world problems are modeled as bandit, across United Nation Refugee Agency assistance program (Caria et al., 2024), government assistance to reduce incarceration (Chohlas-Wood et al., 2023), education app push notification (Yancey & Settles, 2020), news (Li et al., 2010) and movie recommendations (Bibaut et al., 2021).

**Multi-Armed Bandit** In (stochastic) multi-armed bandit problems, we vary our environment configurations primarily along two key dimensions: 1) *action space*, where we change the number of actions $K$ and the textual descriptions associated with each action; 2) *reward distributions*, where we change the parametric distribution of the reward, i.e., the types of reward distributions, and the exploration difficulty, characterized by the gap between the best-performing arm and the second-best arm ($\Delta_{\min}$). A smaller gap makes it harder for the agent to distinguish between optimal and suboptimal actions, thereby increasing the exploration difficulty. In contrast to the setup in Krishnamurthy et al. (2024), which focuses solely on MAB instances with Bernoulli reward distribution, our expanded setup allows us to systematically analyze LLMs' performance across diverse environments

with different action spaces and reward structures. We differentiate between sizes of actions as well as action descriptions: *Videos* such as "Video AA", and *Clothes*, described using semantically meaningful phrases, such as "Supreme Sylvan Sandals". Regarding reward distributions, we evaluate two types: *Bernoulli* and *Gaussian* Bandit. The detailed configurations are shown in Appendix A.2.

**Contextual Bandit** For contextual bandit, at each round $t \in [T]$, the agent is presented with some contextual feature $x$ (which may consist of both textual descriptions and numeric values) describing the state (and action). The LLM agent $\pi$ chooses an action $a \in \mathcal{A}$, and then a reward $r(x, a)$ is received, which depends on both the context and the chosen action. We design the semi-synthetic contextual bandit task based on the MovieLens dataset (Harper & Konstan, 2015), which consists of approximately 6,000 real users' movie ratings. The goal of the agent is to recommend a personalized movie that a specific user is likely to enjoy. In particular, the observations $x$ include user-specific features such as age, gender, occupation, and geographical location (county and state), as well as features of the movies. The action space is limited to the top-$K$ most-watched movies in the dataset, with $K = 10$ for the easy setting and $K = 30$ for the more challenging setting. At each time step, we provide textual contextual features alongside a 5-dimensional user preference vector. The task can easily be scaled up to include more movies (a larger $K$). Further details can be found in Appendix A.3.

## 5. Teaching Optimal Exploration

Motivated by the existence of optimal algorithms for bandits, we aim to leverage these algorithms to improve LLMs for exploration by: 1) incorporating algorithmic guidance during inference (Section 5.1), 2) teaching optimal exploration through algorithm distillation (Section 5.2). We show that smaller models trained using algorithm distillation can even outperform larger models, offering a promising way to efficiently explore with lower inference costs.

Numerous algorithms have been developed to enable efficient exploration in both MAB (Auer, 2002) and CB (Langford & Zhang, 2007; Li et al., 2010) settings. Among these, the Upper Confidence Bound (UCB) algorithm—also known as optimism in the face of uncertainty—stands out for its simplicity and theoretical guarantees. Its clear and interpretable representation of uncertainty and exploration strategy also makes it well-suited for integration with existing LLMs. Our method can generalize to different algorithms easily, such as deep neural network bandits (Riquelme et al., 2018).

**UCB for Multi-Armed Bandit** For MAB, at time step $t$, given the history $\{a_{t'}, r_{t'}\}_{t'=1}^t$, we define $N_t(a)$ as the

number of times that action $a$ has been selected up to time $t$. The empirical mean reward of arm $a$ up to time $t$, denoted as $\hat{\mu}_t(a) := \sum_{t'=1}^{t} \frac{\mathbf{1}_{\{a_{t'}=a\}} r_{t'}}{N_t(a)}$, represents the exploitation value $V^{\text{exploit}}(a, t)$. The high-probability confidence interval, also known as the exploration bonus, is given by $V^{\text{explore}}(a, t) := \alpha \sqrt{\frac{\log(t)}{N_t(a)}}$, where $\alpha$ is the hyperparameter controlling the exploration-exploitation trade-off. At each time step, UCB selects the arm that maximizes the sum of the exploitation value and the exploration bonus, thereby choosing the arm with the highest upper confidence bound.

**UCB for Contextual Bandit**   In CB, we consider the case of linear payoffs (Li et al., 2010; Chu et al., 2011), where the expected reward $\mathbb{E}[r_t^a]$ is assumed to be linear w.r.t a $d$-dimensional feature vector $x_t^a$, with some unknown coefficient vector $\theta^*$, i.e., $\mathbb{E}[r_t^a|x_t^a] = (x_t^a)^T \theta^*$. At each time step, for any arm $a$, the algorithm maintains the design matrix $D_a \in \mathbb{R}^{N_t(a) \times d}$, which represents the feature data for arm $a$ up to time $t$, as well as the corresponding reward vector $r^a \in \mathbb{R}^{N_t(a)}$. It then estimates $\hat{\theta}$ using ridge regression. Moreover, the high-probability confidence interval of the reward estimate $(x_t^a)^T \hat{\theta}$ is given by $\alpha \sqrt{(x_t^a)^T (D_a^T D_a + \lambda I_d)^{-1} x_t^a}$, with $I_d$ being the identity matrix. Following MAB, the exploitation value is the reward estimate, and the exploration bonus is the confidence bound around it.

## 5.1. Inference-Time Algorithm-Guided Support

As discussed above, UCB-type algorithms operate by explicitly calculating the exploitation value $V^{\text{Exploit}}$ along with the exploration bonus $V^{\text{Explore}}$ for each arm, and by selecting the arm that maximizes the sum of the two. These components, $V^{\text{Exploit}}$ and $V^{\text{Explore}}$, therefore provide the sufficient context for LLMs to make optimal decisions. Specifically, in the MAB setup, during inference time at time step $t$, we provide the LLM with a list of tuples $\left(V^{\text{exploit}}(a, t), V^{\text{explore}}(a, t)\right)$ for each arm $a \in [K]$. For CB, during inference-time, we explicitly maintain the design matrix $D_a$ and response vector $r^a$ for each arm, incorporating past interactions from the LLM up to that time $t$, using this to obtain the exploitation value and exploration bonus. We then provide the LLM with a list of exploitation values and exploration bonus for each arm $a$ at current context $x$, similar to the MAB setup. Compared with **SH**, which only provides the empirical mean and the number of times each arm has been pulled, Algorithm-Guided Support (**AG**) directly supplies semantically understandable exploitation values and exploration bonuses. Theoretically, the LLM only needs to perform addition and argmax, rather than manipulating raw histories to discern the underlying reward distribution (or parameter $\theta$ in CB). Another advantage is that **AG** is a type of inference-time support that works seamlessly for both MAB and CB,

while **SH** only works on MAB setup. We discuss why this is the case in Appendix A.5.

## 5.2. Algorithm Distillation

We further investigate the possibility of enhancing LLM exploration by leveraging a set of trajectories generated by an oracle exploration algorithm in the BanditBench environment. This approach, called *algorithm distillation*, aims to distill the optimal exploration behavior from the oracle algorithm to the LLM. In particular, we consider **two approaches**: *in-context few-shot demonstration* and *oracle behavior fine-tuning*, both utilizing expert trajectories generated by the oracle algorithm. Compared with Algorithm-Guided Support (**AG**), these approaches do not require an understanding of the oracle algorithms, nor do they require generating sufficient statistics based on oracle algorithms; thus, they can also be applied to black-box algorithms.

**Oracle Trajectory Generation**   We use UCB as the oracle algorithm to generate the trajectories. Following the notations defined in Section 3, the trajectories are in the form of tuples of $(\phi(H_t^{\text{UCB}}), a_t^{\text{UCB}})$, where each tuple pairs the transformed representation of the history at time $t$ and the action $a_t^{\text{UCB}}$ from UCB. For MAB, we create trajectories from reward distributions that *differ* from those used in evaluation. This assesses the LLM's ability to generalize across different bandit instances with the same underlying scenario but varying *action descriptions* and *action-reward mappings*. We further control the data generation process by varying instance *difficulty* (e.g., $\Delta_{\min}$) and *trajectory contextualization* (e.g., **RH** or **AG**). For CB, we use a fixed dataset and evaluate the LLM's performance on a held-out set of users. While these users are unseen during training, their profiles and preferences remain within the distribution of the training data. In both MAB and CB, each trajectory consists of a sequence of exploration steps: 300 steps for MAB with 5 arms, 1000 steps for MAB with 20 arms, and 200 steps for CB. We generate 50 trajectories for 2 MAB domain configurations (the easiest and the hardest configuration) with 2 trajectory contextualizations, and 200 trajectories for CB with 2 trajectory contextualizations . This results in 4 **distillation datasets** for MAB and 2 for CB.

**Few-Shot Demonstration**   We first study whether demonstrating oracle exploration trajectories from UCB as few-shot examples can improve the LLM's ability to perform robust exploration in bandit tasks. A key challenge in applying few-shot learning to decision-making tasks like MAB is the increasing context length. Unlike supervised learning, where context is typically fixed, bandit actions depend on the entire past history or condensed history, which either grows linearly with $T$ (steps) or $K$ (actions). This poses a challenge for LLMs, as their ability to effectively utilize

information can degrade with longer contexts. We sample 5 oracle trajectories from UCB into the LLM context window as demonstrations. Our goal is to see whether the optimal exploration demonstrations can lead to improved exploration performance. Detailed demonstrations are provided in Appendix A.16.

**Oracle Behavior Fine-Tuning (OFT)** While in-context few-shot demonstrations offers an inference-time approach to guide the LLM's exploration strategy, fine-tuning allows us to directly optimize the model's parameters for the task. In this approach, we utilize the UCB-generated trajectories as training data to adjust the LLM's internal representations and decision-making mechanisms. Specifically, we fine-tune the LLM by framing the exploration problem as a language modeling task, where the goal is to predict the next action in the sequence. This is achieved by maximizing the log-likelihood of the UCB actions given the history of interactions:

$$\mathcal{L}_{\text{OFT}}(\pi) = -\mathbb{E}_{(\phi(H_t^{\text{UCB}}), a_t^{\text{UCB}}) \sim \mathcal{D}_{\text{OFT}}}[\log \pi(a_t^{\text{UCB}} | \phi(H_t^{\text{UCB}}))],$$

where $\pi$ represents the LLM's policy that we aim to optimize. This formulation encourages the LLM to learn the underlying patterns and decision-making logic embedded within the UCB trajectories. By predicting the next action in the sequence, the LLM effectively internalizes the optimal exploration strategy demonstrated by the UCB algorithm. OFT is different from behavior cloning (BC) (Pomerleau, 1991) by learning to encode a dynamic, iterative refinement process, while BC focuses on replicating static behavior of another policy (See Appendix A.7).

# 6. Empirical Evaluations

In this section, we empirically evaluate LLMs' in-context exploration capabilities, using BanditBench. We begin with introducing the setup, baselines and metrics in Section 6.1. Following this, in Section 6.2, we analyze the performance of inference-time guided support, in-context few-shot demonstration and oracle behavior fine-tuning across various experimental settings, as well as models of different sizes. Additionally, we perform extensive ablation studies on the impact of task difficulty, textual representation of the oracle trajectories, and easy-to-hard generalization of the distillation methods. We also analyze the types of failures encountered by LLMs in in-context exploration, thereby demonstrating the necessity of developing more advanced algorithms.

## 6.1. Setup

**Baselines** We evaluate the in-context exploration capabilities of various LLMs: Gemma-2B, Gemma-9B (Team et al., 2024), Gemini 1.5 Flash, and Gemini 1.5 Pro (Reid et al., 2024), on 16 MAB tasks (Table A1) and 2 CB tasks. For

MAB tasks, the interaction horizon ($T$) differs based on the size of the action space ($K$): we use $T = 1000$ for $K = 30$ and $T = 200$ for $K = 10$. All CB tasks use a constant horizon of 200 steps. To ensure statistical significance of the results, we conduct 30 independent runs for each experimental setup. We consider two baselines: Raw History (**RH**) and Summarized History (**SH**), as suggested in Krishnamurthy et al. (2024). For CB, as we discussed before, there is no trivial analogue of **SH**; thus, we focus solely on **RH** for CB tasks in this study as the baseline.

**Metrics** We report the relative performance of each model, aggregated across all environment configurations. Simply averaging cumulative rewards across environments of different reward distributions and horizons however, obscures the comparison. We instead use the *pairwise* win-rate to compare the performances. We have 16 configurations for MAB and evaluate 32 models and 2 configurations for CB with 14 models. The list of all the models is provided in Appendix A.14. For each configuration, we compute the cumulative reward over $T$ steps and collect their distribution from 30 independent trials. We then calculate the pairwise win-rate by applying a Student's $t$-test on the reward distributions of any pair of configurations to determine if they are statistically significantly different, with a significance level of $p < 0.05$. The *overall win-rate* for a model is then the percentage of superior performance over all models, crossed with methods and configurations. Details are given in Appendix A.8.

## 6.2. Results and Ablations

**Overall Performance Comparison** Figure 1 presents a comparative overview of in-context few-shot demonstration, oracle behavior fine-tuning, and inference-time algorithmic guidance performance across various model sizes and training configurations. Detailed numbers are reported in Table A4 and A5. Few-shot demonstrations exhibited contrasting effects on Gemini-1.5 Flash and Pro. While few-shot learning boosts the performance of Flash beyond the best history contextualization setup, it hurts Pro's performance in both MAB and CB. Aligned with the observations in Zheng et al. (2024); Guo et al. (2025), our hypothesis is that few shot examples we manually crafted could disrupt the CoT structure in larger models, which requires the few-shot examples to be carefully tuned in order to be helpful. Further analysis reveals the remarkable effectiveness of oracle behavior fine-tuning. It significantly outperforms both few-shot and baseline approaches in both MAB and CB across all model sizes, even larger ones. This robust improvement highlights the effectiveness of directly optimizing model parameters for the exploration task. Notably, the best fine-tuned Gemini-1.5 Flash model surpasses even the Gemini-1.5 Pro model, highlights its potential as a key technique for enhancing LLM exploration capabilities.

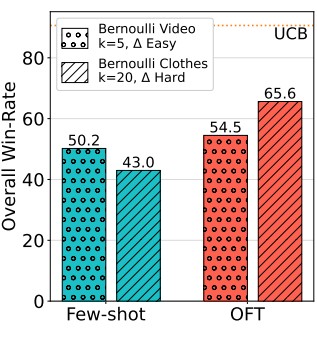

(a) Task Difficulty (MAB).

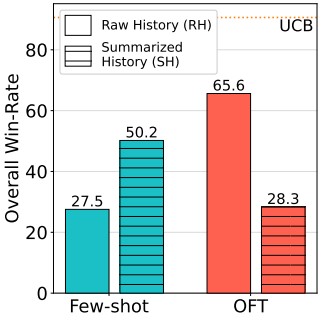

(b) History Representation, **RH** vs **SH** (MAB).

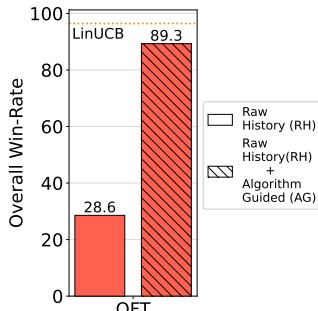

(c) History Representation with and without **AG** (CB).

Figure 2: **Impact of task difficulty and history representation on algorithm distillation.** This figure examines how different factors, such as task difficulty and history representation of oracle trajectories, influence the effectiveness of algorithm distillation on the LLM's exploration capabilities. All results are based on Gemini-1.5 Flash model. The numbers represent the best model in that setting. Same number indicates the same model.

|            | Flash (**RH**) | Flash (**AG**) | Pro (**RH**) | Pro (**AG**) |
|------------|:--------------:|:--------------:|:------------:|:------------:|
| MAB, K=5   | **33.6**       | 26.6           | 48.0         | **67.6**     |
| MAB, K=20  | 21.9           | **37.9**       | 43.0         | **51.6**     |
| CB, K=10   | 0.0            | **35.7**       | 7.1          | **57.1**     |
| CB, K=30   | 0.0            | **57.1**       | 7.1          | **71.4**     |

Table 1: Situations where Algorithm-Guided Support significantly outperforms Raw History during inference.

**Algorithm-Guided Support Helps on Complex Environments**   We observe consistent improvements when transitioning from **RH** to **AG** in two scenarios: (1) larger models like Flash and Pro, and (2) more complex scenarios, such as contextual bandits. Table 1 shows that **AG** provided consistent help when the number of actions is large for both Flash and Pro models. We hypothesize that providing **AG** is crucial when the action space is large or when decision scenarios are complex.

**Impact of Task Difficulty**   We examine whether the choice of oracle trajectories used in both few-shot demonstration and oracle behavior fine-tuning affects the model's performance during inference. To investigate this, we select trajectories from two extreme setups. The easiest setup involves *(Bernoulli, Video, Large $\Delta_{min}$, $K = 5$)*, denoted as $D_{easy}$, with **AG**. Conversely, the hardest setup, denoted as $D_{hard}$ utilizes *(Bernoulli, Clothes, Small $\Delta_{min}$, $K = 20$)*, with **RH**. Figure 2a illustrates that the choice of oracle trajectories significantly impacts the model's performance, with a surprising contrast between the two algorithm distillation methods. Few-shot demonstration achieves a higher win-rate when using $D_{easy}$ as demonstration (50.2) compared to when using $D_{hard}$ (43.0). This suggests that the limited examples provided in the demonstrations may be insufficient for the model to effectively utilize them under the higher complexity and subtle reward signals of the harder task.

Conversely, fine-tuning exhibits the opposite trend, with a higher win-rate when trained on $D_{hard}$ (65.6) compared to $D_{easy}$ (54.5). This implies that fine-tuning, with its extensive training data, might be overfitting to the specific nuances of the training distribution, leading to poor generalization when faced with a different task structure.

**Impact of Contextualization**   We further investigate the effect of contextualization in oracle trajectories. We consider two representations in MAB: **RH** and **SH**. Results in Figure 2b reveal a clear contrast in how these representations affect the two algorithm distillation methods. For few-shot demonstration, **SH** leads to significantly better performance (50.2% win-rate) compared to **RH** (27.5% win-rate). This suggests that providing concise, informative summaries of optimal exploration behavior is more effective for few-shot learning than presenting the complete raw history. On the other hand, fine-tuning exhibits the opposite trend. **RH** has a substantially higher win-rate (65.6) compared to **SH** (28.3). This indicates that fine-tuning benefits from the richer information present in complete action-reward sequences, allowing it to learn more nuanced patterns of the optimal exploration strategy. These contrasting preferences for textual representation in oracle trajectories highlight the nuanced ways in which fine-tuning and few-shot learning interact with different types of information. Furthermore, in CB, we observe a significant impact of incorporating **AG** information into the oracle trajectories for fine-tuning, leading a dramatic improvement in win-rate, rising from 28.6 to 89.3 (Figure 2c). This suggests that providing LLMs with explicit insights, in addition to the complete action-reward sequence, enhances its ability to learn and generalize the optimal exploration strategy in the CB environment.

**Algorithm Distillation Generalizes**   We also conducted experiments to evaluate the distillation algorithm's ability to generalize from smaller to larger action spaces (i.e., easy-to-

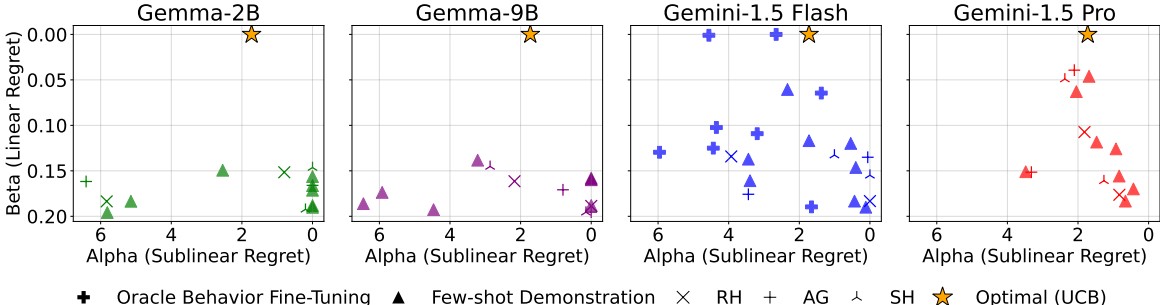

Figure 3: Bernoulli **MAB** in Hard ($K$=20, $\Delta_{\min}$=0.2): We plot the estimated parameters $\alpha$ and $\beta$. Smaller $\alpha$ and $\beta$ indicate more efficient exploration to find the best arm. Algorithms with strong in-context exploration should have $\alpha$ as small as possible and have $\beta$=0. We can see for MAB Hard setup, lesser models achieved sublinear regret. We show the rest of MAB setups in Table A5, A6.

hard domain generalization). This analysis offers insights into the scalability and adaptability of our approach to more complex domains, such as real-world recommendation systems and other complex decision-making problems. We report this in Table 2. Using few-shot demonstrations or doing oracle behavior fine-tuning from simple-domain trajectories collected (Bernoulli, Easy, K=5) can indeed learn exploration strategies that generalize to a harder domain (Bernoulli, Hard, K=20).

| MAB Hard | Flash (**SH**) | Flash + Few-Shot (**SH**) ($\mathcal{D}_{\text{easy}}$ on K=5) | Flash + OFT (**RH**) ($\mathcal{D}_{\text{easy}}$ on K=5) |
|---|---|---|---|
| K=20 | 34.0% | 43.0% | 48.4% |

Table 2: **Easy-to-Hard Generalization**: We collect distillation data from easy bandit tasks to learn the exploration behavior and evaluate on hard bandit tasks. History representation during evaluation is in parentheses.

**Optimality Analysis of Exploration** We use two metrics to capture high-level exploration behaviors: (1) **MinFrac**, proposed by Krishnamurthy et al. (2024), captures the fraction of pulls allocated to the least-selected arm. An ideal algorithm should exhibit high **MinFrac** during early exploration and gets lower as $t$ increases, indicating effective exploration; (2) **OptFrac**, tracks the percentage of times the optimal arm is pulled. Ideally, this percentage should increase as the process progresses, indicating the model's ability to self-improve. Using Flash model and Bernoulli bandit configurations as an example in Table 3, we see that with **AG**, the Flash model on average explores more compared to **SH** and have higher success at identifying the best arm.

While a model might achieve high performance by chance—e.g., consistently selecting the optimal arm without deliberate exploration—we examine its behavior more closely using two metrics. **OptFrac** measures how often the optimal arm is chosen, revealing whether the model increasingly favors the best option. As shown in Table 3, UCB's

OptFrac steadily rises over time (32.7% $\rightarrow$ 65.0%), while Gemini-1.5 Flash remains nearly flat (9.3% $\rightarrow$ 10.7%), suggesting limited adaptation. To assess directed exploration, we use **MinFrac**, the fraction of pulls allocated to the least-selected arm. UCB shows a desirable decline (82.3% $\rightarrow$ 15.3%), indicating early broad exploration followed by exploitation. In contrast, Gemini-1.5 Flash starts low and drops quickly (11.3% $\rightarrow$ 1.1%), implying minimal directed exploration throughout. Together, these metrics demonstrate that Gemini's performance is not driven by meaningful exploration. More discussion in Appendix A.11.

| **MinFrac** / $t$ Step | 100-th | 250-th | 500-th | 750-th | 1000-th |
|---|---|---|---|---|---|
| UCB | 82.3% | 48.6% | 27.8% | 19.6% | 15.3% |
| Flash (**SH**) | 10.2% | 4.2% | 2.1% | 1.4% | 1.1% |
| Flash (**AG**) | 11.3% | 4.5% | 2.3% | 1.5% | 1.1% |
| **OptFrac** | | | | | |
| UCB | 32.7% | 49.4% | 58.7% | 62.6% | 65.0% |
| Flash (**SH**) | 9.3% | 10.1% | 10.4% | 10.6% | 10.7% |
| Flash (**AG**) | 14.4% | 15.6% | 16.3% | 16.6% | 16.8% |

Table 3: **Optimality of Exploration**: We show the measures on the $t$-th step along the progression of exploration, averaged across Bernoulli MAB configurations. Full version with more models in Table A6 and A7.

## 7. Regret Analysis of LLM Exploration

In this section, we propose a **new** and **more rigorous** analysis of the LLM's exploration efficiency using the concept of regret, $REG(\pi)$. Most bandit algorithms are evaluated by the behavior of $REG(\pi)$ as a function of $T$ (i.e., the number of interactions), either theoretically or empirically. Motivated by this, our goal is to understand the exploration behaviors of various LLMs by characterizing their regret as a function of $T$. We adopt the following functional form to analyze the regret: $f(T) = \frac{\lambda_1 \log(T)^{\alpha}}{\Delta_{\min}} + \beta T + \lambda_2$. The parameters $\alpha, \beta, \lambda_1$ in the equation are all positive real numbers. $\lambda_2$ is unconstrained. This functional form provides intuitive interpretations for the underlying parameters. Specifically, $\alpha$ represents sublinear scaling of the regret, which

is known to be achieved by only the best bandit algorithms (e.g. UCB). The $\beta$ scaling describes a linear growth or the inability of an agent to match the optimal policy $\pi^*$. This means a strong algorithm should have $\alpha$ as small as possible, and have $\beta = 0$. This functional form also allows us to see some growth behaviors in-between with both positive $\alpha$ and $\beta$. Details of fitting is in Appendix A.12. We use the curve fit function in Scikit-learn (Pedregosa et al., 2011) to fit the cumulative regret curve of UCB and LLMs coupled with different methods. The results of the fitted $\alpha$ and $\beta$ values are presented in Figure 3. For the largest Pro models, applying effective inference-time support, such as **AG** and **SH** can achieve nearly sub-linear regret. More intriguingly, for Flash models, fine-tuning for optimal behavior significantly boosts performance, enabling them to attain sub-linear regret with a lower $\alpha$. In contrast, weaker models such as Gemma 2B and 9B appear to remain in the linear regret regime across nearly all methods.

## 8. Conclusion

In this work, we explored the in-context exploration capabilities of LLMs in bandit environments, introducing Bandit-Bench, a comprehensive benchmark designed to rigorously evaluate LLM's performance. Our evaluation reveals that LLMs struggle with in-context exploration when relying solely on raw interaction history, while inference-time support significantly improves performance. Motivated by the presence of optimal algorithms in this domain, we investigated methods to integrate these algorithms into LLMs through both algorithm-guided support and algorithm distillation via synthesized demonstration data. Notably, these approaches enable smaller models to outperform larger ones in decision-making tasks. However, an optimality gap remains between LLMs and classical optimal algorithms, highlighting the need for further research to bridge this gap.

## Software and Data

BanditBench and the inference code have been provided in this GitHub repo and will be updated/monitored regularly: https://github.com/allenanie/EVOLvE. You can install the code with: `pip install banditbench`.

## Acknowledgements

We thank Yaqing Wang for helping some infra on data processing and storage, Aviral Kumar, Yifeng Lu, Haokai Lu, Ice Pasupat, Kefan Dong, Ollie Liu, Deqing Fu for valuable discussions on the project. We thank Ruijie Zheng for feedback on the code base.

## Impact Statement

Our goals are two fold – first, using the bandit framework to understand whether LLMs have the ability to make effective decisions given context, which is distinct from their ability to reason and plan. Second, we want to explore whether well-known methods (such as few-shot demonstrations and supervised fine-tuning) can teach LLMs how to explore more optimally. There are many potential societal consequences of our work, but they are outside the scope of our paper.

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

# A. Appendix

**Multi-Armed Bandit (Clothes)**

You are an AI fashion assistant for an on-line boutique powered by a bandit algorithm that offers a variety of clothing options from different brands.
[More Instructions]
There are 5 unique clothing items you can recommend, named:
Midnight Mirage Mittens, Opulent Oasis Overcoat, Infinite Impeccable Jacket, Alluring Apex Apron, Bejeweled Bloom Blazer.
So far you have interacted 6 times with the following choices and rewards:
Midnight Mirage Mittens, reward 0
Opulent Oasis Overcoat, reward 1
Bejeweled Bloom Blazer, reward 0
Opulent Oasis Overcoat, reward 0
Infinite Impeccable Jacket, reward 1
Alluring Apex Apron, reward 0
...
Which item will you choose next?

**Contextual Bandit (Movies)**

You are an AI movie recommendation assistant for a streaming platform powered by a bandit algorithm that offers a wide variety of films from different studios and genres.
[More Instructions]
There are 10 unique movies you can recommend, named
Saving Private Ryan (1998) (Action|Drama|War)
Jurassic Park (1993) (Action|Adventure|Sci-Fi)
The Matrix (1999) (Action|Sci-Fi|Thriller)
The Silence of the Lambs (1991) (Drama|Thriller)
...
So far you have interacted 4 times with the following choices and rewards:
Context: This person is an 18-year-old college/grad student living in Pulaski county, AR. The user has some numerical values that represent their true implicit preference or true taste for all movies: [-0.011, 0.027, -0.020, -0.002, -0.003].
Action: Saving Private Ryan (1998)
Reward: 4.74 out of 5
...
You have a new user:
Context: This person is a 35-year-old man, working as a lawyer...
Action:

Figure A1: The problem representation of in-context exploration is represented in text. Detailed prompts for both MAB and CB are provided in Appendix A.15.

**Raw History**

[Scenario Description]
[Instructions]
[List of Actions]
Past Raw History:
Time 1, Action Name, reward $r_1$
Time 2, Action Name, reward $r_2$
Time 3, Action Name, reward $r_3$
Time 4, Action Name, reward $r_4$
...
Which [Action] will you choose next?

**Summarized History with Algorithm-Guided Support**

[Scenario Description]
[Instructions]
[List of Actions]
Summarized History:
Action 1 Name, chosen $n$ times, average reward $\hat{\mu}^1$, exploration bonus $v_1$, exploitation bonus $e_1$.
Action 2 Name, chosen $n$ times, average reward $\hat{\mu}^2$, exploration bonus $v_2$, exploitation bonus $e_2$..
...
Which [Action] will you choose next?

Figure A2: The problem representation of in-context exploration in text. For Summarized History (**SH**), the text in gray is presented. For Algorithm-Guided Support (**AG**), the text in pink and yellow are presented along with the text in gray. This schema works for any *general* algorithm that explicitly compute exploration and exploitation bonus. For UCB, $e_1 = \hat{\mu}^1$. Detailed prompts for both MAB and CB are provided in Appendix A.15.

## A.1. Extended Related Work

Several prior works have investigated the use of LLMs for decision-making. In one category, numerous studies have deployed LLMs directly as agents in decision-making problems such as games (Yao et al., 2023; Brooks et al., 2024; Shinn et al., 2024; Wang et al., 2023a; Xi et al., 2023). However, fewer works have systematically evaluated LLMs' capabilities in general decision-making setup, especially in relation to classical concepts in decision-making like exploration. Our work extends the research of Krishnamurthy et al. (2024), who examined LLMs' exploration capabilities in small-scale MAB tasks. Their findings, which showed positive results only with substantial intervention, are consistent with our broader analysis across both MAB and CB tasks at various scales. Mirchandani et al. (2023); Rahn et al. (2024); Felicioni et al. (2024) also evaluated the ability of LLMs to learn in-context and solve bandit-like decision-making problems.

The line of research on using LLMs as optimizers faces many similar challenges to in-context decision making, although

applied to different tasks. Yang et al. (2024) explored the use of language models as general-purpose optimizers for simple black-box optimization problems, such as prompt optimization, highlighting that a careful balance of exploration and exploitation is critical. Another relevant line of research focuses on in-context learning for decision-making and reinforcement learning (RL) with domain-specific transformers. Laskin et al. (2022) distilled demonstrations from RL algorithms into a transformer and showed that the transformer learns to imitate the RL process to solve new RL tasks. Similarly, Lee et al. (2024) trained transformers with optimal action labels, showing that the model learns to execute posterior sampling for RL (Osband et al., 2013) in-context, which tailors exploration to the underlying distribution of RL tasks. This area has been further studied by Raparthy et al. (2023); Lin et al. (2023). However, these studies focus on domain-specific decision-making, whereas our paper examines general-purpose decision-making capabilities in language models.

Our inference-time algorithm-guided support shares a similar conceptual framework with recent efforts to align LLMs at inference time. These include employing explicit value functions as prefix scorers that trained via KL-regularized RL (Mudgal et al., 2023), and leveraging both implicit and explicit value functions to guide decoding at the token and chunk levels at inference time (Liu et al., 2024b). In the realm of algorithm distillation, much of the research on LLMs has concentrated on chain-of-thought (CoT) reasoning (Wang et al., 2023b; Li et al., 2023b), while (Gandhi et al., 2024) focused on search and backtracking. Most methods involve distilling outputs from a "teacher" model—either a larger model or a slower, system-2 variant of the same model that employs various inference-time techniques, such as search and self-consistency—into a student model (Yu et al., 2024). Instead, our approach leverages diverse optimal trajectories directly from classical algorithms, allowing for the efficient generation of abundant training data.

### A.2. Details on Multi-Armed Bandit Task

In (stochastic) multi-armed bandit problems, we vary our environment configurations primarily along two key dimensions: 1) *action space*, where we change the number of actions $K$ and the textual descriptions associated with each action; 2) *reward distributions*, where we change the parametric distribution of the reward, i.e., the types of reward distributions, and the exploration difficulty, characterized by the gap between the best-performing arm and the second-best arm. A smaller gap makes it harder for the agent to distinguish between optimal and sub-optimal actions, thereby increasing the exploration difficulty. In contrast to the setup in Krishnamurthy et al. (2024), which focuses solely on MAB instances with Bernoulli reward distribution, our expanded setup allows us to systematically analyze LLMs' performance across diverse environments with different action spaces and reward structures.

For the action space, we explore two different sizes: $K = 5$ for a small action space and $K = 20$ for a large action space. We also differentiate between two types of action descriptions: *Videos*, represented as arbitrary two-letter combinations with no semantic meaning such as "Video AA", and *Clothes*, described using semantically meaningful phrases, such as "Supreme Sylvan Sandals". Regarding reward distributions, we evaluate two types: *Bernoulli* and *Gaussian* Bandit. For Bernoulli, the reward $r \in \{0, 1\}$ is binary with $r^{a_k} \sim \text{Bernoulli}(p_k)$, where $p_k$ is the mean for the $k$-th action. Following Krishnamurthy et al. (2024), the best-performing arm has $p_k := 0.5 + \Delta_{\min}/2$, while the remaining arms have $p_k = 0.5 - \Delta_{\min}/2.0$ The parameter $\Delta_{\min}$ captures the exploration difficulty, with a larger gap ($\Delta_{\min} = 0.5$) indicating easy tasks and smaller gap ($\Delta_{\min} = 0.2$) representing hard tasks. For the Gaussian bandit, the rewards are continuous with $r^{a_k} \sim \mathcal{N}(\mu_k, \sigma)$. Here $\mu_k \sim \mathcal{N}(0, \sigma)$ represents the mean for each action, and the variance $\sigma$ captures the difficulty of exploration. Following Sutton (2018), we study both $\sigma = 1$ and $\sigma = 3$.

We have 16 configurations for the multi-armed bandit domain, shown in Table A1.

| | Parameters | |
|---|---|---|
| Reward Type | Bernoulli | Gaussian |
| Exploration Difficulty | Easy ($\Delta_{\min}$=0.5), Hard ($\Delta_{\min}$=0.2) | Easy ($\sigma = 1$), Hard ($\sigma = 3$) |
| Number of Items/Actions | Small ($k = 5$), Large ($k = 20$) | |
| Action Description | Videos, Clothes | |

Table A1: Configuration of the MAB setup.

### A.3. Details on Contextual Bandit Task

For contextual bandit, at each round $t \in [T]$, the agent is presented with some contextual feature $x$ (which may consist of both textual descriptions and numeric values) describing the state (and action). The LLM agent $\pi$ chooses an action $a \in \mathcal{A}$, and then a reward $r(x, a)$ is received, which depends on both the context and the chosen action. The goal of the agent is to recommend a personalized movie that a specific user is likely to enjoy. In particular, the observations $x$ include user-specific features such as age, gender, occupation, and geographical location (county and state), as well as features of the movies. The action space is limited to the top-$K$ most-watched movies in the dataset, with $K = 10$ for the easy setting and $K = 30$ for the more challenging setting. At each time step, we provide textual contextual features alongside a 5-dimensional user preference vector $u_i$. The task can easily be scaled up to include more movies, i.e., a larger $K$.

We use the MovieLens-1M dataset (Harper & Konstan, 2015) to build the contextual bandit task. It contains 1,000,209 anonymous ratings of approximately 3,900 movies made by 6,040 MovieLens users who joined MovieLens in 2000. For each user, we have the basic demographic information such as age, gender, occupation, and zip code. We further convert zip code to the actual name of the county and state and add these into the user profile description text. Each movie has a title and associated genres. We present these information in the prompt as well.

LinUCB assumes a linear reward model $\mathbb{E}[r|x, a] = \theta_a^T x$, where $\theta \in \mathbb{R}^d$ (Chu et al., 2011). Since we are trying to use tasks to measure the performance of LLM against a theoretically optimal algorithm, we have to build the contextual bandit task in a way that satisfies the LinUCB assumption. An additional issue is that the context window of an LLM is still limited, and we want to restrict the number of movies for the LLM to 10 or 30. So, we first calculate the popular movies by tracking how often users rate each. We sort the list and select the top $K$ movies. Then, we build a user preference matrix $P \in \mathbb{R}^{N \times K}$, where $N$ is the number of users and $K$ is the number of movies. To construct the ground-truth reward distribution, we perform low-rank approximation on $P$. This is done by approximating $P$ with $\tilde{P} = U\Sigma V^T$ using singular value decomposition (SVD), yielding a user embedding matrix $U \in \mathbb{R}^{N \times d}$, a movie embedding matrix $V \in \mathbb{R}^{K \times d}$, and a diagonal matrix $\Sigma \in \mathbb{R}^{d \times d}$ of the top singular values. In our case, we set $d = 5$ as the dimension of the embeddings. The ground-truth reward for user $i$ and movie $j$ is then computed as $r_{i,j} = u_i^T \Sigma v_j$.

In order to present the full information that was provided to LinUCB to LLM as well, we include the user preference vector in the prompt space, represented by a list of 5 floating point numbers. We additionally add descriptions to indicate that this is a user preference vector. We show our full prompt in Figure A13.

### A.4. UCB and LinUCB

In Table A2, we provide a detailed comparison between the exploitation values and exploration bonus used in both UCB and LinUCB.

| Algorithm | Task | Value of Arm |
|---|---|---|
| UCB | MAB | $V_t(a) = \underbrace{\hat{\mu}_t(a)}_{V^{\text{Exploit}}} + \underbrace{\alpha\sqrt{\log(t)/N_t(a)}}_{V^{\text{Explore}}}$ |
| LinUCB | CB | $V_t(a, x) = \underbrace{x_{t,a}^T \hat{\theta}_a}_{V^{\text{Exploit}}} + \underbrace{\alpha\sqrt{x_{t,a}^T(D_a^T D_a + I_d)^{-1} x_{t,a}}}_{V^{\text{Explore}}}$ |

Table A2: Calculation for the value of each arm/item. The decision rule is $a^* = \arg\max_a V_t(a, x)$.

### A.5. History Representation in Contextual Bandit

While it is relatively clear-cut for multi-armed bandit what **RH** and **SH** correspond to, it is less so for contextual bandit. If we were to perform a similar analysis with LinUCB, **RH** would correspond to retaining all (context, action, reward) information to estimate the parameter and calculate the uncertainty, while one possibility to realize **SH** would be to construct the sufficient statistics using running mean and running covariance matrix in LinUCB. However, these statistics are much less interpretable for language models; thus, we do not investigate it.

### A.6. Details on Oracle Trajectory Generation

We use UCB as the oracle algorithm to generate the trajectories. Following the notations defined in Section 3, the trajectories are in the form of tuples of $(\phi(H_t^{\text{UCB}}), a_t^{\text{UCB}})$, where each tuple pairs the transformed representation of the history at time $t$

and the action $a_t^{\text{UCB}}$ from UCB. For MAB, we create trajectories from reward distributions that *differ* from those used in evaluation. This assesses the LLM's ability to generalize across different bandit instances with the same underlying scenario but varying *action descriptions* and *action-reward mappings*. We further control the data generation process by varying: (1). *Action Description*: trajectories are generated from either "Video" or "Clothes" action descriptions; (2). *Difficulty*: we control the reward gap in the Bernoulli bandit to create "easy" and "hard" instances; (3). *Trajectory Contextualization*: trajectories are represented either as **RH** or **AG**. For CB, we use a fixed dataset and evaluate the LLM's performance on a held-out set of users. While these users are unseen during training, their profiles and preferences remain within the distribution of the training data. This evaluates the LLM's ability to leverage prior knowledge for effective exploration. In CB, we only vary the trajectory representation (**RH** or **AG**). In both MAB and CB, each trajectory consists of a sequence of exploration steps: 300 steps for MAB with $K = 5$ arms, 1000 steps for MAB with $K = 20$ arms, and 200 steps for CB. We generate 50 trajectories for 2 MAB domain configurations (the easiest and the hardest configuration) with 2 trajectory contextualizations, and 200 trajectories for CB with 2 trajectory contextualizations . This results in 4 algorithm distillation datasets for MAB and 2 datasets for CB.

### A.7. Difference Between Algorithm Distillation and Behavior Cloning

Optimal Behavior Fine-tuning (OFT) and Behavior Cloning (Pomerleau, 1991) share many similarities. Although both approaches rely on maximum-likelihood learning, their objectives are different: OFT seeks to encode a dynamic, iterative refinement process, while BC focuses on replicating static behavior. OFT is designed for algorithm distillation, focusing on capturing a sequence of self-improvement behaviors, and generalization across any new test domains. In contrast, BC aims to learn a policy by mimicking a static policy, with no iterative improvement between trajectories.

This difference becomes very clear when we think of an example. We have a deterministic Markov policy $\pi$ that we can use to create this dataset. We call this the sampling policy. To create a behavior cloning dataset, $\mathcal{D}_{\text{BC}}$, during dataset construction, for the same state $s$, the policy remains unchanged, which the means $\pi(a|s)$ remains the same in the entire dataset. To create an algorithm distillation dataset $\mathcal{D}_{\text{OFT}}$, the sampling policy is self-improving as the data collection continues, $\pi(a|s)$ changes even for the same $s$ between early and late trajectories of this dataset.

### A.8. Example of Win-Rate Calculation

We report the relative performance of each model, aggregated across all environment configurations. Simply averaging cumulative rewards across environments of different reward distributions and horizons however, obscures the comparison. We instead use the *pairwise* win-rate to compare the performances. We have 16 configurations for MAB and evaluate 32 models (4 LLMs crossed with different methods), and 2 configurations for CB with 14 models (2 LLMs crossed with different methods). The list of all the models is provided in Appendix A.14. For each configuration, we compute the cumulative reward over $T$ steps and collect a distribution of cumulative rewards from 30 independent trials. We then calculate the pairwise win-rate by applying a Student's $t$-test on the reward distributions of any pair of configurations to determine if they are statistically significantly different, with a significance level of $p < 0.05$. If one model has a significantly higher reward than the other, we consider it a win. If the difference is not statistically significant, the result is deemed inconclusive and not counted as a win. For each model, we calculate its win rate against every other model across all configurations. The *overall win-rate* for a model is then the percentage of superior performance over all models, crossed with methods and configurations.

In each configuration, we compute one model's win-rate against all other models. For MAB, we have 16 configurations and 34 models. For CB, we have 2 configurations and 16 models. Finally, the model's *overall win-rate* is then determined by averaging its win-rates across all models. For example, in MAB, if we only have 3 models: Gemma-2B, Gemini-1.5 Flash, and Pro. Gemini-1.5 Flash have higher expected cumulative reward than Gemma-2B in 12 out of 16 configurations (12/16), but only higher than Gemini-1.5 Pro in 4 out of 16 configurations (4/16), Gemini-Flash 1.5 will have an overall win-rate, on average, 8/16=0.5.

### A.9. Benchmark Evaluation Cost

We calculate and report the inference time cost for evaluating 30 trials for one agent with the longest contextual representation (which incurs the highest cost). This means, for MAB tasks, we evaluate using Raw History (**RH**) and for CB tasks, we evaluate using Algorithm-Guided Support (**AG**). The total cost for MAB Full (over 16 configurations and 32 models) is $15897.6 for Gemini-1.5 Flash and for CB (2 configurations and 14 models) is $14491.12 for Gemini-1.5 Flash. We will release all logged experiment data of all models for public analysis and comparison. We also recommend using a subset of

|                    | Core     | HardCore | HardCore+ | Full      | MovieBench |
|--------------------|----------|----------|-----------|-----------|------------|
| Gemini-1.5 Flash   | $31.05   | $14.91   | $39.18    | $83.44    | $31.05     |
| Gemini-1.5 Pro     | $517.54  | $248.55  | $652.98   | $1390.69  | $517.54    |
| GPT-4o             | $1035.07 | $497.11  | $1305.96  | $2781.38  | $1035.07   |
| Claude-3.5-sonnet  | $1243.00 | $596.91  | $1567.91  | $3339.53  | $1243.00   |

Table A3: Maximal inference cost for a single agent/task configuration over 30 trials. Price calculated on 1/30/2025. Core, HardCore and HardCore+ refer to specific subset of the MAB configurations. The exact details are in the code included in the supplementary material. In this paper, we used Full (MAB) and MovieBench (CB) of the benchmark.

our configurations for MAB (HardCore and HardCore+) to reduce evaluation cost.

### A.10. Details Discussion of Results

**Overall Performance Comparison**    Figure 1 presents a comparative overview of in-context few-shot demonstration, oracle behavior fine-tuning, and inference-time algorithmic guidance performance across various model sizes and training configurations. Few-shot demonstrations exhibited contrasting effects on Gemini-1.5 Flash and Pro. While few-shot learning boosts the performance of Flash beyond the best inference-time setup, it surprisingly hurts Pro's performance in both MAB and CB. Aligned with the observations in Zheng et al. (2024), our hypothesis is that few shot examples we manually crafted could disrupt the CoT structure in these larger models, which requires the few-shot examples to be carefully tuned in order to be helpful. Further analysis reveals the remarkable effectiveness of oracle behavior fine-tuning. It significantly outperforms both few-shot and baseline approaches in both MAB and CB across all model sizes, even larger ones. This robust improvement highlights the effectiveness of directly optimizing model parameters for the exploration task. Notably, the best fine-tuned Gemini-1.5 Flash model surpasses even the highest-performing Gemini-1.5 Pro model. The significant advantage of fine-tuning over few-shot learning and baseline performance highlights its potential as a key technique for enhancing LLM exploration capabilities.

**Impact of History Representation in Context**    We examine how different contextualizations and inference-time support techniques—namely **RH**, **SH**, and **AG**—influence model performance, each utilizing distinct history contextualization functions $\phi$, as introduced in Section 3. It is worth mentioning that in the MAB setup, both **SH** and **AG** significantly reduce context length compared to **RH**, to $O(K)$ instead of $O(t)$. As illustrated in Table A4, leveraging inference-time support (i.e., **SH** and **AG**) significantly enhances exploration performance across all models. This supports the intuition that effective in-context exploration requires more than memorizing input-output pairs; it demands reasoning to extract sufficient statistics from raw data and utilize them effectively for balancing exploration and exploitation. However, the exact benefit of incorporating UCB-style information in the MAB setup remains uncertain. We hypothesize that under MAB, the exploitation value and exploration bonus are straightforward transformations of the empirical mean and the number of times each arm has been pulled $N_t(a)$, and that the LLM has the capacity to learn the functional form efficiently. In CB, we compare **AG** to **RH** and find a substantial improvement. This gap is particularly significant, as learning the exploitation value and exploration bonus in this scenario requires the model to implicitly solve ridge regression and determine the appropriate functional form of the high-probability confidence bound, making it a more complex reasoning task. The algorithmic guide approach can thus be seen as LLMs calling external tools to compute sufficient statistics required for optimal exploration.

| Inference-Time Support | Multi-Armed Bandit | | | | Contextual Bandit | |
|---|---|---|---|---|---|---|
|  | Gemma-2B | Gemma-9B | Flash | Pro | Flash | Pro |
| Raw History (**RH**) | 7.6 | 10.5 | 27.7 | 45.5 | 0.0 | 7.1 |
| Summarized History (**SH**) | 10.5 | 5.3 | 34.8 | 60.0 | – | – |
| Algorithm-Guided Support (**AG**) | 4.9 | 4.1 | 32.2 | 59.6 | 46.4 | 64.3 |
| UCB / LinUCB | | 90.6 | | | 96.4 | |

Table A4: **Inference-Time Support Results**: Overall Win-Rate (%) of different inference support. Flash and Pro refer to Gemini-1.5 Flash and Pro respectively. Unlike **SH**, **AG** can work for both MAB and CB. Refer to Section 5.1 for details.

| (a) **Few-shot Demonstrations** | | | | |
| --- | --- | --- | --- | --- |
| | **MAB** | | **CB** | |
| | Flash | Pro | Flash | Pro |
| Raw History (**RH**) | 27.5 | 39.1 | 3.6 | 7.1 |
| Algorithm-Guided Support (**AG**) | **50.2** | **56.4** | **60.7** | **25.0** |

| (b) **Oracle Behavior Fine-Tuning** | | | | |
| --- | --- | --- | --- | --- |
| | **MAB** | | **CB** | |
| | Flash | Pro | Flash | Pro |
| Raw History (**RH**) | **65.6** | — | 28.6 | — |
| Algorithm-Guided Support (**AG**) | 28.3 | — | **89.3** | — |

Table A5: **Algorithm Distillation Results**: Overall Win-Rate (%) of different algorithm distillation methods. Flash and Pro refer to Gemini-1.5 Flash and Pro respectively. Best achieved performances are in bold. The history contextualization used in oracle trajectory and inference-time support are the same. We conduct a few ablations in Figure A3.

**Impact of Task Difficulty in Distillation Datasets**  We examine whether the choice of oracle trajectories used in both in-context demonstration and oracle behavior fine-tuning significantly affects the model's performance during inference. To investigate this, we select trajectories from two extreme setups. The easiest setup involves *(Bernoulli, Video, Large* $\Delta_{min}$, $K = 5$), denoted as $D_{easy}$, with **AG**. Conversely, the hardest setup, denoted as $D_{hard}$ utilizes *(Bernoulli, Clothes, Small* $\Delta_{min}$, $K = 20$), with **RH**. Figure 2a illustrates that the choice of oracle trajectories significantly impacts the model's performance, with a surprising contrast between the two algorithm distillation methods. In-context demonstration achieves a higher win-rate when using $D_{easy}$ as demonstration (50.2) compared to when using $D_{hard}$ (43.0). This suggests that the limited examples provided in the demonstrations may be insufficient for the model to effectively utilize them under the higher complexity and subtle reward signals of the harder task. Conversely, fine-tuning exhibits the opposite trend, with a higher win-rate when trained on $D_{hard}$ (65.6) compared to $D_{easy}$ (54.5). This implies that fine-tuning, with its extensive training data, might be overfitting to the specific nuances of the training distribution, leading to poor generalization when faced with a different task structure.

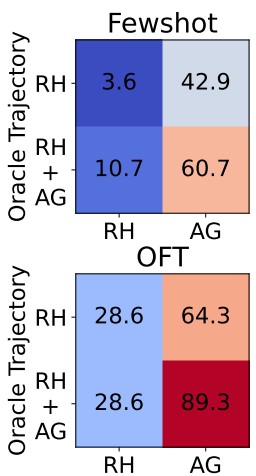

Figure A3: Context Representation Alignment.

**Impact of Contextualization**  We further investigate the effect of contextualization in the oracle trajectories. We consider two representations in MAB: **RH** and **SH**. The results in Figure 2b reveal a clear contrast in how these representations affect the two algorithm distillation methods. For few-shot demonstration, **SH** leads to significantly better performance (50.2% win-rate) compared to **RH** (27.5% win-rate). This suggests that providing concise, informative summaries of optimal exploration behavior is more effective for few-shot learning than presenting the complete raw history. On the other hand, fine-tuning exhibits the opposite trend. **RH** has a substantially higher win-rate (65.6) compared to **SH** (28.3). This indicates that fine-tuning benefits from the richer information present in complete action-reward sequences, allowing it to learn more nuanced patterns of the optimal exploration strategy. These contrasting preferences for textual representation in oracle trajectories highlight the nuanced ways in which fine-tuning and few-shot learning interact with different types of information. Furthermore, in CB, we observe a significant impact of incorporating algorithm-guided (**AG**) information into the oracle trajectories for fine-tuning. Augmenting **RH** with **AG** details, including the exploitation value and exploration bonus, leads to a dramatic improvement in win-rate, rising from 28.6 to 89.3 in Figure 2c. This suggests that providing the LLM with explicit insights into the underlying decision-making process of the oracle algorithm (UCB, in this case), in addition to the complete action-reward sequence, significantly enhances its ability to learn and generalize the optimal exploration strategy in the CB environment.

**Impact of Context Representation Alignment**  Our experiments also reveal an interesting interplay between the presence of algorithm-guided information (**AG**) in both the oracle *trajectories* and *inference*. In the CB setting, providing **AG** during inference consistently boosts performance, regardless of whether **AG** was used in oracle trajectories. This is clearly demonstrated in Figure A3, where the right column (with **AG** at inference time) exhibits higher win-rates than the corresponding left column across all training conditions. This suggests that the LLM can effectively leverage this information even if it wasn't explicitly trained on it, highlighting the inherent value of structured guidance for decision-making. Furthermore, we observe that incorporating **AG** into few-shot demonstrations improves exploration even when **AG** is absent during inference (e.g., Fewshot, **RH** 3.6 to **RH** +**AG** 10.7). This indicates that exposing the LLM to **AG** in oracle trajectories, even in a limited capacity, can enhance its ability to extract relevant patterns from **RH**. We hypothesize that **AG** helps the LLM learn to focus on the most informative aspects of the history, which generalizes even when **AG** is not

provided during inference.

## A.11. Details on Exploration Optimality Analysis

There are two types of failures one can expect in a bandit problem:

1. **Over-exploration on suboptimal choices which results in lower exploration efficiency**: over-exploration happens when the algorithm spends too much time exploring suboptimal choices, reducing overall efficiency. This behavior can be quantified using the **MinFrac** metric (Krishnamurthy et al., 2024), which measures the fraction of pulls allocated to the least-selected arm. An ideal algorithm should exhibit high **MinFrac** during early exploration (when $T$ is small) and low **MinFrac** as $T$ increases (indicating effective exploitation).

2. **Failure to identify the optimal arm**: this occurs when the algorithm struggles to converge on the best option over time. To capture this, we compute the percentage of times an optimal arm is pulled at different time steps (**OptFrac**). Ideally, this percentage should increase as the process progresses, indicating the model's ability to self-improve.

We report these metrics at specific time steps over a total of T time steps. For convenience in visualizing the results in a table, we select the 10%, 25%, 50%, 75%, and 100% (final) time steps. There are 1000 steps for the Bernoulli The reported numbers are from Bernoulli Bandit, the metrics are averaged across different configurations.

| MinFrac (%) / $t$ Step | 100-th | 250-th | 500-th | 750-th | 1000-th |
|---|---|---|---|---|---|
| UCB | 82.3 | 48.6 | 27.8 | 19.6 | 15.3 |
| Gemma-2B (**SH**) | 0.0 | 1.0 | 0.5 | 0.4 | 0.3 |
| Gemma-2B (**AG**) | 0.0 | 0.6 | 1.1 | 1.9 | 2.7 |
| Gemma-9B (**SH**) | 6.9 | 11.2 | 16.9 | 15.4 | 16.5 |
| Gemma-9B (**AG**) | 5.8 | 11.8 | 17.3 | 19.6 | 22.9 |
| Gemini-1.5 Flash (**SH**) | 10.2 | 4.2 | 2.1 | 1.4 | 1.1 |
| Gemini-1.5 Flash (**AG**) | 11.3 | 4.5 | 2.3 | 1.5 | 1.1 |
| Gemini-1.5 Pro (**SH**) | 79.0 | 40.0 | 20.6 | 13.9 | 10.5 |
| Gemini-1.5 Pro (**AG**) | 73.8 | 37.1 | 18.9 | 12.7 | 9.5 |

Table A6: **Over-exploration Rate (MinFrac)**: MinFrac measures the fraction of pulls allocated to the least-selected arm. We show the measures on the $t$-th step along the progression of exploration.

| OptFrac (%) / $t$ Step | 100-th | 250-th | 500-th | 750-th | 1000-th |
|---|---|---|---|---|---|
| UCB | 32.7 | 49.4 | 58.7 | 62.6 | 65.0 |
| Gemma-2B (**SH**) | 10.1 | 10.3 | 10.2 | 10.1 | 10.1 |
| Gemma-2B (**AG**) | 12.8 | 12.3 | 12.4 | 12.2 | 12.2 |
| Gemma-9B (**SH**) | 5.8 | 6.7 | 7.4 | 7.8 | 8.0 |
| Gemma-9B (**AG**) | 6.6 | 7.3 | 6.7 | 6.6 | 6.6 |
| Gemini-1.5 Flash (**SH**) | 9.3 | 10.1 | 10.4 | 10.6 | 10.7 |
| Gemini-1.5 Flash (**AG**) | 14.4 | 15.6 | 16.3 | 16.6 | 16.8 |
| Gemini-1.5 Pro (**SH**) | 15.1 | 19.1 | 21.8 | 22.9 | 23.5 |
| Gemini-1.5 Pro (**AG**) | 11.9 | 17.7 | 21.6 | 23.1 | 23.9 |

Table A7: **Fraction of Pulls on the Optimal Arm (OptFrac)**: OptFrac measures the fraction of pulls overall on the optimal arm over. We show the measures on the $t$-th step along the progression of exploration.

A model could, in theory, achieve high performance by consistently choosing the optimal arm—even if it rarely explores—by chance. To address this, we include an analysis of the model's exploration behavior using a metric called **OptFrac**, which measures how often the optimal arms are selected. As shown in Table A7, UCB steadily increases its OptFrac over time ($32.7\% \rightarrow 49.4\% \rightarrow 58.7\% \rightarrow 62.6\% \rightarrow 65.0\%$ over 1000 steps), indicating a growing focus on the optimal arm. In contrast, Gemini-1.5 Flash remains largely flat ($9.3\% \rightarrow 10.1\% \rightarrow 10.4\% \rightarrow 10.6\% \rightarrow 10.7\%$), suggesting that it does not significantly shift its behavior toward the optimal arm. This supports our claim that the model does not accidentally achieve optimal performance by randomly selecting the best arm without meaningful exploration.

We then analyze exploration dynamics in more detail, such as whether the model engages in random or directed exploration. In our analysis, we include a metric called MinFrac, which measures the fraction of pulls allocated to the least-selected arm. This captures the extent to which the model explores less-visited options, which can be understood as a form of "directed exploration." Ideally, this value should be high early on (indicating strong directed exploration), and then decrease as the model gains experience and focuses on better-performing arms. As shown in Table A6, UCB exhibits this expected trend, with MinFrac values decreasing over time: $82.3\% \rightarrow 48.6\% \rightarrow 27.8\% \rightarrow 19.6\% \rightarrow 15.3\%$. In contrast, Gemini-1.5 Flash starts with a much lower MinFrac and declines rapidly ($11.3\% \rightarrow 4.5\% \rightarrow 2.3\% \rightarrow 1.5\% \rightarrow 1.1\%$), suggesting it lacks meaningful directed exploration from the outset.

### A.12. Details on Fitting Regret Function

In this section, we aim to conduct a more rigorous analysis of the LLM's exploration efficiency using the concept of regret, $REG(\pi)$. Most bandit algorithms are evaluated by the behavior of $REG(\pi)$ as a function of $T$ (i.e., the number of interactions), either theoretically or empirically. Motivated by this, our goal is to understand the exploration behaviors of various LLMs by characterizing their regret as a function of $T$. To achieve this, we adopt the following functional form to analyze the regret:

$$f(T) = \frac{\lambda_1 \log(T)^\alpha}{\Delta_{\min}} + \beta T + \lambda_2$$

The three parameters $\alpha, \beta, \lambda_1$ in the equation are all positive real numbers. $\lambda_2$ is unconstrained. $\Delta_{\min}$ captures the gap between the best and second best arm. This functional form provides intuitive interpretations for the underlying parameters. Specifically, $\log(T)$ represents sublinear scaling of the regret, which is known to be achieved by only the best bandit algorithms (e.g. UCB and Thompson Sampling). The $T$ scaling describes a linear growth or the inability of an agent to match the optimal policy $\pi^*$. This means a strong algorithm should have $\alpha$ as small as possible, and have $\beta = 0$. This functional form also allows us to see some growth behaviors in-between with both positive $\alpha$ and $\beta$.

We use the curve fit function in Scikit-learn (Pedregosa et al., 2011) to fit the cumulative regret curve of UCB and LLMs coupled with different methods (i.e., inference-time algorithm-guided support, in-context demonstration, and optimal behavior fine-tuning). The results of the fitted $\alpha$ and $\beta$ values are presented in Figure 3. For the largest Pro models, applying effective inference-time support, such as **AG** and **SH** can achieve nearly sub-linear regret. More intriguingly, for Flash models, fine-tuning for optimal behavior significantly boosts performance, enabling them to attain sub-linear regret with a lower $\alpha$. In contrast, weaker models such as Gemma 2B and 9B appear to remain in the linear regret regime across nearly all methods.

We perform the same analysis with the cumulative regret function on MAB in the Hard Difficulty setting. We can see that in Figure 3, a lot fewer LLM models achieved $\beta = 0$, which means achieving the desirable logarithmic sublinear regret that algorithms like UCB and Thompson Sampling have.

In the MAB-Hard setting, we can see that more models are having non-zero $\beta$, meaning these models have linear cumulative regret, which indicates lack of in-context self-improvement, because the model is not selecting the optimal arm more and more frequently as $T$ increases. However, we can see that generally Optimal Behavior Fine-Tuned models are doing better.

To verify that our functional form fits the data (empirical cumulative regret curve) well, we show a few figures of our fitted curve and actual data. In Figure A4, we show how the learned function $f(T)$ fit the actual empirical cumulative regret curve.

In Figure A4, it is interesting to see that the function we choose exhibit the behavior of pushing either $\alpha$ or $\beta$ to 0, if either of the two describes the trend better. We note that although the fit is not perfect, the MSE is relatively small compared to the data we are trying to fit. For a cumulative regret as large as 100 at some time step $T$, our fitted function ccan still maintain an MSE of 0.22.

We additionally add the analysis for the MAB Gaussian bandit. The instance optimality gap $\Delta_{\min}$ is characterized by the KL-divergence of the Gaussian reward distribution between the best arm and the second best arm. We show the result in Figure A6. The trend is somewhat similar to Bernoulli bandits, where smaller models perform much worse than larger models.

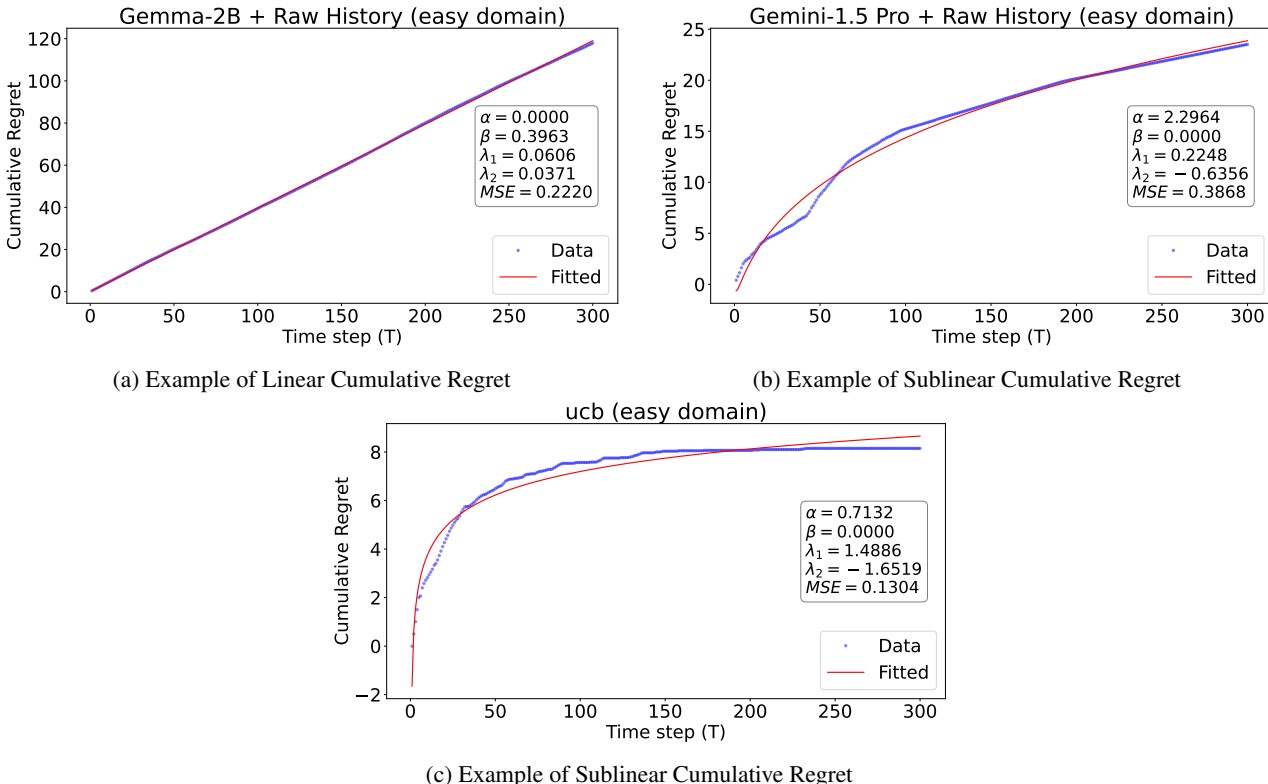

(a) Example of Linear Cumulative Regret

(b) Example of Sublinear Cumulative Regret

(c) Example of Sublinear Cumulative Regret

Figure A4: Examples of how our function fits different empirical cumulative regret curves. $T$ indicates number of times the agent interacted with the task.

### A.13. Evaluation Implementation Details

We run each model under each configuration for 30 trials. We set the random seed to be the same as trial id, starting from 0 to 29. This random seed determines the reward distribution for MAB and the sequence of users the algorithm encounters in CB. For the LLM calls, we use standard API calls and set the sampling temperature to 1.0 (range=[0.0, 2.0]). The default API (2024-08 to 2024-09) uses Top-P=0.95 sampling, and Top-K=40.

### A.14. Full List of Models

We provide a full list of models evaluated for MAB and CB. The model is represented using A $\implies$ B with A being the model, with B being the inference-time technique.

**MAB Models**

1. Few-Shot Gemma-9B, (Bernoulli, Clothes, $K = 20$, Small $\Delta_{min}$) $\implies$ **RH**     0.029

2. Few-Shot Gemma-2B, (Bernoulli, Clothes, $K = 20$, Small $\Delta_{min}$) $\implies$ **RH**     0.029

3. Gemma-9B $\implies$ **AG**     0.041

4. Fewshot Gemma-2B with (Bernoulli, Video, $K = 5$, Large $\Delta_{\min}$) $\implies$ **SH**     0.043

5. Fewshot Gemma-2B with (Bernoulli, Clothes, $K = 20$, Small $\Delta_{min}$) $\implies$ **SH**     0.045

6. Fewshot Gemma-2B with (Bernoulli, Video, $K = 5$, Large $\Delta_{\min}$) $\implies$ **RH**     0.047

7. Gemma-2B $\implies$ **AG**     0.049

8. Gemma-9B $\implies$ **SH**     0.053

9. Fewshot Gemma-9B with (Bernoulli, Video, $K = 5$, Large $\Delta_{\min}$) $\implies$ **RH**     0.072

10. Gemma-2B $\implies$ **RH**     0.076

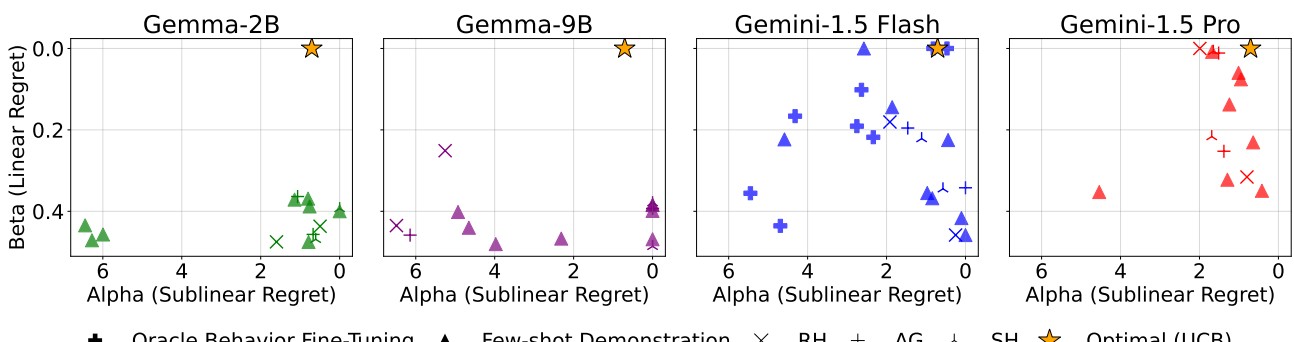

Figure A5: **MAB** in Easy ($K$=5, $\Delta_{\min}$=0.5): We plot the estimated parameters $\alpha$ and $\beta$. Smaller $\alpha$ and $\beta$ indicate more efficient exploration to find the best arm. Algorithms with strong in-context exploration should have $\alpha$ as small as possible and have $\beta$=0.

11. Fewshot Gemma-9B with (Bernoulli, Clothes, $K = 20$, Small $\Delta_{min}$) $\implies$ **SH**    0.088

12. Fewshot Gemma-9B with (Bernoulli, Video, $K = 5$, Large $\Delta_{\min}$) $\implies$ **SH**    0.092

13. OFT Flash with (Bernoulli, Video, $K = 5$, Large $\Delta_{\min}$) **AG** $\implies$ **AG**    0.104

14. Gemma-2B $\implies$ **SH**    0.105

15. Gemma-9B $\implies$ **RH**    0.105

16. Fewshot Flash with (Bernoulli, Clothes, $K = 20$, Small $\Delta_{min}$) $\implies$ **RH**    0.152

17. Fewshot Flash with (Bernoulli, Video, $K = 5$, Large $\Delta_{\min}$) $\implies$ **RH**    0.275

18. Gemini-1.5 Flash $\implies$ **RH**    0.277

19. OFT Flash with (Bernoulli, Clothes, $K = 20$, Small $\Delta_{min}$) **AG** $\implies$ **AG**    0.283

20. Gemini-1.5 Flash $\implies$ **AG**    0.322

21. Gemini-1.5 Flash $\implies$ **SH**    0.348

22. Fewshot Pro with (Bernoulli, Video, $K = 5$, Large $\Delta_{\min}$) $\implies$ **RH**    0.381

23. Fewshot Pro with (Bernoulli, Clothes, $K = 20$, Small $\Delta_{min}$) $\implies$ **RH**    0.391

24. Fewshot Flash with (Bernoulli, Clothes, $K = 20$, Small $\Delta_{min}$) $\implies$ **SH**    0.430

25. Gemini-1.5 Pro $\implies$ **RH**    0.455

26. Fewshot Flash with (Bernoulli, Video, $K = 5$, Large $\Delta_{\min}$) $\implies$ **SH**    0.502

27. Fewshot Pro with (Bernoulli, Clothes, $K = 20$, Small $\Delta_{min}$) $\implies$ **SH**    0.525

28. OFT Flash with (Bernoulli, Video, $K = 5$, Large $\Delta_{\min}$) **RH** $\implies$ **RH**    0.545

29. Fewshot Pro with (Bernoulli, Video, $K = 5$, Large $\Delta_{\min}$) $\implies$ **SH**    0.564

30. Gemini-1.5 Pro $\implies$ **AG**    0.596

31. Gemini-1.5 Pro $\implies$ **SH**    0.600

32. OFT Flash with (Bernoulli, Clothes, $K = 20$, Small $\Delta_{min}$) **RH** $\implies$ **RH**    0.656

33. UCB    0.906

**CB Models**

1. Gemini-1.5 Flash $\implies$ **RH**    0.000

2. Fewshot Flash with **RH** $\implies$ **RH**    0.036

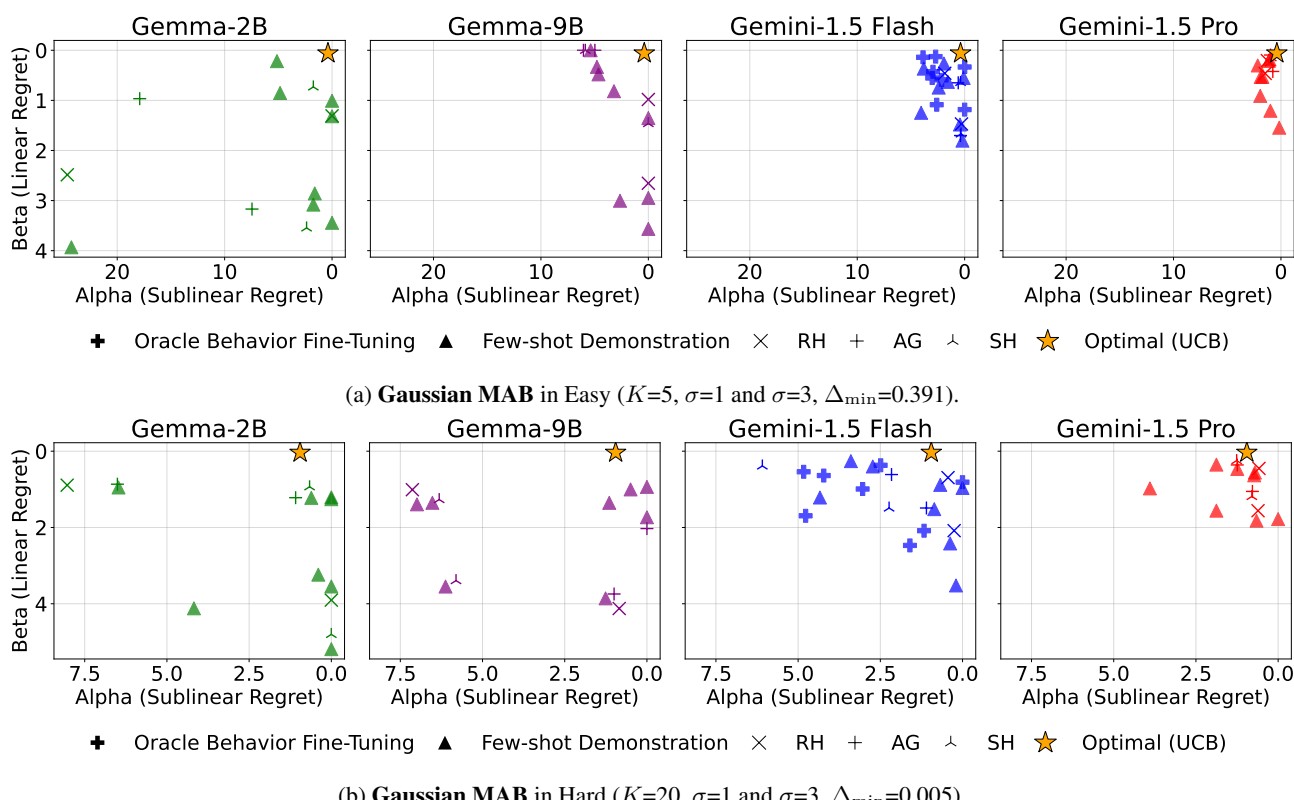

(a) **Gaussian MAB** in Easy ($K$=5, $\sigma$=1 and $\sigma$=3, $\Delta_{\min}$=0.391).

(b) **Gaussian MAB** in Hard ($K$=20, $\sigma$=1 and $\sigma$=3, $\Delta_{\min}$=0.005).

Figure A6: We plot the estimated parameters $\alpha$ and $\beta$. The difficulty level is determined by $\Delta_{\min}$.

3. Fewshot Pro with **RH** $\implies$ **RH**                                    0.071

4. Gemini-1.5 Pro $\implies$ **RH**                                             0.071

5. Fewshot Flash with **RH** $\implies$ **RH**                                  0.107

6. Fewshot Pro with **RH** $\implies$ **AG**                                    0.250

7. OFT trained with **RH** $\implies$ **RH**                                    0.286

8. OFT trained with **AG** $\implies$ **RH**                                    0.286

9. Fewshot Flash with **RH** $\implies$ **AG**                                  0.429

10. Gemini-1.5 Flash $\implies$ **AG**                                          0.464

11. Fewshot Flash with **AG** $\implies$ **AG**                                 0.607

12. OFT trained with **RH** $\implies$ **AG**                                   0.643

13. Gemini-1.5 Pro $\implies$ **AG**                                            0.643

14. OFT trained with **AG** $\implies$ **AG**                                   0.893

15. LinUCB                                                                      0.964

## A.15. Scenario Prompts

We provide a set of prompts that are used in each scenario. For Multi-Arm Bandit, we include the following prompts:

1. MAB, Bernoulli Bandit, $K = 5$, Raw History (**RH**), Video Action Description (Figure A7), Clothes Action Description (Figure A8)

2. MAB, Bernoulli Bandit, $K = 5$, Algorithm-Guided Support (**AG**), Clothes Action Description (Figure A9), Video Action Description (Figure A10)

3. MAB, Gaussian Bandit, $K = 5$, Raw History (**RH**), Video Action Description (Figure A11), Clothes Action Description (Figure A12)

For Contextual Bandit, we include the following prompts:

1. CB, $K = 10$, Raw History (**RH**) (Figure A13)

2. CB, $K = 10$, Raw History (**RH**) with Algorithm-Guided Support (**AG**) (Prompt Part 1 Figure A14, Prompt Part 2 Figure A15).

For **OFT**, we use the same prompt as shown in the figures above. The LLM generates the next action token conditioned on the entire prompt, and we compute the negative log-likelihood loss over the action tokens, with the action chosen by UCB/LinUCB algorithm.

### A.16. Examples of few-shot demonstrations

We provide examples of how few-shot prompt being used. We include few-shot demonstrations from optimal exploration trajectories before past interaction history (without the task description and instruction). We show two examples to illustrate that how few-shot demonstrations domain match with the evaluation domain:

1. MAB, Benoulli Bandit, Video Action Description, $K = 5$, Raw History (**RH**), with Few-shot Demonstrations from Video Action Description, $K = 5$, Raw History (**RH**) (Figure A16)

2. MAB, Benoulli Bandit, Video Action Description, $K = 5$, Raw History (**RH**), ith Few-shot Demonstrations from Clothes Action Description, $K = 5$, Raw History (**RH**) (Figure A17)

```
1    You are a video recommendation system powered by a bandit algorithm for an online streaming platform.
2    There are 5 videos available in your library, titled [A, B, AI, BS, E].
3    When a user logs into the platform, you select a video to recommend based on their viewing history and
      preferences.
4    You aim to engage the user by recommending videos that they are likely to watch.
5    Each time a user watches a recommended video, you update your recommendation model to refine future
      suggestions,
6    enhancing user satisfaction and platform engagement.
7
8    A good strategy to optimize for reward in these situations requires balancing exploration
9    and exploitation. You need to explore to try out all of the videos and find those
10   with high rewards, but you also have to exploit the information that you have to
11   accumulate rewards.
12
13   So far you have played 6 times with the following choices and rewards:
14   A video, reward 1
15   B video, reward 1
16   AI video, reward 1
17   BS video, reward 0
18   E video, reward 0
19   A video, reward 0
20
21   Which video will you choose next? PLEASE RESPOND ONLY WITH A, B, AI, BS, E AND NO TEXT EXPLANATION.
22
```

Figure A7: Multi-Arm Bandit: Bernoulli, Video Action Description, $K = 5$, Raw History.

```
1    You are an AI fashion assistant for an online boutique powered by a bandit algorithm that offers a variety of
       clothing options from different brands.
2    There are 5 unique clothing items you can recommend, named [Midnight Mirage Trousers, Opulent Oasis Overcoat,
       Infinite Impeccable Jacket, Supreme Spectrum Slippers, Bejeweled Bloom Blazer].
3    When a customer visits the online store, you assess their style preferences and shopping history to choose an
       item to suggest.
4    You aim to match the customer with clothing they are most likely to purchase and enjoy.
5    Each time a customer buys a recommended item, you adjust your recommendation algorithms to better predict and
       meet future customer preferences.
6
7    A good strategy to optimize for reward in these situations requires balancing exploration
8    and exploitation. You need to explore to try out all of the clothing brands and find those
9    with high rewards, but you also have to exploit the information that you have to
10   accumulate rewards.
11
12   So far you have played 6 times with the following choices and rewards:
13   Midnight Mirage Trousers item, reward 0
14   Opulent Oasis Overcoat item, reward 1
15   Infinite Impeccable Jacket item, reward 1
16   Supreme Spectrum Slippers item, reward 0
17   Bejeweled Bloom Blazer item, reward 0
18   Opulent Oasis Overcoat item, reward 1
19
20   Which item will you choose next? PLEASE RESPOND ONLY WITH Midnight Mirage Trousers, Opulent Oasis Overcoat,
       Infinite Impeccable Jacket, Supreme Spectrum Slippers, Bejeweled Bloom Blazer AND NO TEXT EXPLANATION.
21
```

Figure A8: Multi-Arm Bandit: Bernoulli, Clothing Action Description, $K = 5$, Raw History.

```
1    You are an AI fashion assistant for an online boutique that offers a variety of clothing options from
       different brands.
2    There are 5 unique clothing items you can recommend, named
3    Stellar Sheen Shawl,
4    Faithful Fantasy Frock,
5    Supreme Sylvan Sandals,
6    Bespoke Bliss Blouse item,
7    Silk Spectrum Slip
8    When a customer visits the online store, you assess their style preferences and shopping history to choose an
       item to suggest.
9    You aim to match the customer with clothing they are most likely to purchase and enjoy.
10   Each time a customer buys a recommended item, you adjust your recommendation algorithms to better predict and
       meet future customer preferences.
11   A good strategy to optimize for reward in these situations requires balancing exploration
12   and exploitation. You need to explore to try out all of the clothing brands and find those
13   with high rewards, but you also have to exploit the information that you have to
14   accumulate rewards.
15   So far you have played 4 times with the following choices and rewards:
16   Stellar Sheen Shawl item, 1 time, avg reward 0, exploration bonus 1.00, exploitation value 0.00
17   Faithful Fantasy Frock item, 1 time, avg reward 1, exploration bonus 1.00, exploitation value 1.00
18   Supreme Sylvan Sandals item, 1 time, avg reward 0, exploration bonus 1.00, exploitation value 0.00
19   Bespoke Bliss Blouse item, avg reward 0, exploration bonus 1.00, exploitation value 0.00
20   Silk Spectrum Slip item, 1 time, avg reward 0, exploration bonus 1.00, exploitation value 0.00
21   Which clothes item will you choose next?
22   Action:
23
```

Figure A9: Multi-Arm Bandit: Bernoulli, Clothing Action Description, $K = 5$, Algorithmic Guide.

```
1   You are a video recommendation system powered by a bandit algorithm for an online streaming platform.
2   There are 5 videos available in your library, titled
3   AA
4   BS
5   BW
6   CQ
7   CP
8   When a user logs into the platform, you select a video to recommend based on their viewing history and
     preferences.
9   You aim to engage the user by recommending videos that they are likely to watch.
10  Each time a user watches a recommended video, you update your recommendation model to refine future
     suggestions, enhancing user satisfaction and platform engagement.
11  A good strategy to optimize for reward in these situations requires balancing exploration
12  and exploitation. You need to explore to try out all of the videos and find those
13  with high rewards, but you also have to exploit the information that you have to
14  accumulate rewards.
15  So far you have played 4 times with the following choices and rewards:
16  AA video, 1 time, avg reward 0, exploration bonus 1.00, exploitation value 0.00
17  BS video, 1 time, avg reward 1, exploration bonus 1.00, exploitation value 1.00
18  BW video, 1 time, avg reward 0, exploration bonus 1.00, exploitation value 0.00
19  CQ video, avg reward 0, exploration bonus 1.00, exploitation value 0.00
20  CP video, 1 time, avg reward 0, exploration bonus 1.00, exploitation value 0.00
21  Which video will you choose next?
22  Action:
23
```

Figure A10: Multi-Arm Bandit: Beroulli, Video Action Description, $K = 5$, Algorithmic Guide.

```
1   You are a video recommendation system powered by a bandit algorithm for an online streaming platform.
2   There are 5 videos available in your library, titled [A, CX, AF, AQ, S].
3   When a user logs into the platform, you select a video to recommend based on their viewing history and
     preferences.
4   You aim to engage the user by recommending videos that they are likely to watch.
5   Each time a user watches a recommended video, you update your recommendation model to refine future
     suggestions,
6   enhancing user satisfaction and platform engagement.
7
8   A good strategy to optimize for reward in these situations requires balancing exploration
9   and exploitation. You need to explore to try out all of the videos and find those
10  with high rewards, but you also have to exploit the information that you have to
11  accumulate rewards.
12
13  So far you have played 6 times with the following choices and rewards:
14  A video, reward 2.0205556227286694
15  CX video, reward 5.046038662976072
16  AF video, reward -4.043037070451992
17  AQ video, reward 5.937910707405409
18  S video, reward -4.856036829535051
19  AQ video, reward 6.2468398842187405
20
21  Which video will you choose next? PLEASE RESPOND ONLY WITH A, CX, AF, AQ, S AND NO TEXT EXPLANATION.
22
```

Figure A11: Multi-Arm Bandit: Gaussian, Video Action Description, $K = 5$, Raw History.

```
1    You are an AI fashion assistant for an online boutique powered by a bandit algorithm that offers a variety of
      clothing options from different brands.
2    There are 5 unique clothing items you can recommend, named [Midnight Mirage Trousers, Dapper Dreams Denim,
      Infinite Impeccable Jacket, Supreme Spectrum Slippers, Bejeweled Bloom Blazer].
3    When a customer visits the online store, you assess their style preferences and shopping history to choose an
      item to suggest.
4    You aim to match the customer with clothing they are most likely to purchase and enjoy.
5    Each time a customer buys a recommended item, you adjust your recommendation algorithms to better predict and
      meet future customer preferences.
6
7    A good strategy to optimize for reward in these situations requires balancing exploration
8    and exploitation. You need to explore to try out all of the clothing brands and find those
9    with high rewards, but you also have to exploit the information that you have to
10   accumulate rewards.
11
12   So far you have played 6 times with the following choices and rewards:
13   Midnight Mirage Trousers item, reward -3.701605707528312
14   Dapper Dreams Denim item, reward 1.4965799995904072
15   Infinite Impeccable Jacket item, reward 4.576557137862691
16   Supreme Spectrum Slippers item, reward -0.32883145604929176
17   Bejeweled Bloom Blazer item, reward 1.5907554114707747
18   Infinite Impeccable Jacket item, reward 6.534020380965033
19
20   Which item will you choose next? PLEASE RESPOND ONLY WITH Midnight Mirage Trousers, Dapper Dreams Denim,
      Infinite Impeccable Jacket, Supreme Spectrum Slippers, Bejeweled Bloom Blazer AND NO TEXT EXPLANATION.
21
```

Figure A12: Multi-Arm Bandit: Gaussian, Clothes Action Description, $K = 5$, Raw History.

```
1 You are an AI movie recommendation assistant for a streaming platform powered by a bandit algorithm that offers a
      wide variety of films from different studios and genres.
2 There are 10 unique movies you can recommend, named
3 American Beauty (1999) (Comedy|Drama),
4 Star Wars: Episode IV - A New Hope (1977) (Action|Adventure|Fantasy|Sci-Fi),
5 Star Wars: Episode V - The Empire Strikes Back (1980) (Action|Adventure|Drama|Sci-Fi|War),
6 Star Wars: Episode VI - Return of the Jedi (1983) (Action|Adventure|Romance|Sci-Fi|War),
7 Jurassic Park (1993) (Action|Adventure|Sci-Fi),
8 Saving Private Ryan (1998) (Action|Drama|War),
9 Terminator 2: Judgment Day (1991) (Action|Sci-Fi|Thriller),
10 The Matrix (1999) (Action|Sci-Fi|Thriller),
11 Back to the Future (1985) (Comedy|Sci-Fi),
12 The Silence of the Lambs (1991) (Drama|Thriller)
13
14 When a user visits the streaming platform, you assess their demographic description to choose a movie to suggest.
15 You aim to match the user with movies they are most likely to watch and enjoy.
16 Each time a user watches a recommended movie, you adjust your recommendation algorithms to better predict and
      meet future user preferences.
17 Your goal is to enhance the user's viewing experience by providing personalized and engaging movie suggestions.
18
19 A good strategy to optimize for reward in these situations requires balancing exploration
20 and exploitation. You need to explore to try out different movies and find those
21 with high rewards, but you also have to exploit the information that you have to
22 accumulate rewards.
23
24 So far you have interacted 4 times with the most recent following choices and rewards:
25 Context: a person who is a 18-year-old man with an occupation of college/grad student and live in Pulaski county,
      AR. The user has some numerical values that represent their true implicit preference or taste for all
      movies: [-0.011492758058011532, 0.027099572122097015, -0.020118921995162964, -0.002230832353234291,
      -0.003236030228435993].
26 Action: Saving Private Ryan (1998)
27 Reward: 4.735634 out of 5
28
29 Context: a person who is a 25-year-old man with an occupation of sales/marketing and live in Solano county, CA.
      The user has some numerical values that represent their true implicit preference or taste for all movies:
      [-0.00312434253282845, 0.0017211971571668983, 0.0015880014980211854, 0.012064018286764622,
      0.009061760269105434].
30 Action: Jurassic Park (1993)
31 Reward: 0 out of 5
32
33 Context: a person who is a 56-year-old man with an occupation of sales/marketing and live in Jefferson county,
      KY. The user has some numerical values that represent their true implicit preference or taste for all
      movies: [-0.009686884470283985, 0.028794225305318832, -0.011435767635703087, 0.006439171731472015,
      -0.010343835689127445].
34 Action: Saving Private Ryan (1998)
35 Reward: 5 out of 5
36
37 Context: a person who is a 25-year-old man with an occupation of executive/managerial and live in Washington
      county, DC. The user has some numerical values that represent their true implicit preference or taste for
      all movies: [-0.010095382109284401, 0.010144174098968506, -0.01811344549059868, -0.009553882293403149,
      -0.012143188156187534].
38 Action: Saving Private Ryan (1998)
39 Reward: 3.953174 out of 5
40
41
42 You have a new user: PLEASE RESPOND ONLY WITH A CHOICE of MOVIES LISTED ABOVE AND NO TEXT EXPLANATION.
43
44 Context: This person is a 35-year-old man, working as a lawyer and live in Camden county, NJ. The user has some
      numerical values that represent their true implicit preference or taste for all movies:
      [-0.009149148128926754, -0.00417252816259861, 0.011747784912586212, -0.012008273974061012,
      -0.006486567202955484].
45 Action:
46
```

Figure A13: Contextual Bandit: Movie Recommendation for movies, Raw History.

```
1 You are an AI movie recommendation assistant for a streaming platform powered by a bandit algorithm that offers a
      wide variety of films from different studios and genres.
2 There are 10 unique movies you can recommend, named
3 American Beauty (1999) (Comedy|Drama),
4 Star Wars: Episode IV - A New Hope (1977) (Action|Adventure|Fantasy|Sci-Fi),
5 Star Wars: Episode V - The Empire Strikes Back (1980) (Action|Adventure|Drama|Sci-Fi|War),
6 Star Wars: Episode VI - Return of the Jedi (1983) (Action|Adventure|Romance|Sci-Fi|War),
7 Jurassic Park (1993) (Action|Adventure|Sci-Fi),
8 Saving Private Ryan (1998) (Action|Drama|War),
9 Terminator 2: Judgment Day (1991) (Action|Sci-Fi|Thriller),
10 The Matrix (1999) (Action|Sci-Fi|Thriller),
11 Back to the Future (1985) (Comedy|Sci-Fi),
12 The Silence of the Lambs (1991) (Drama|Thriller)
13
14 When a user visits the streaming platform, you assess their demographic description to choose a movie to suggest.
15 You aim to match the user with movies they are most likely to watch and enjoy.
16 Each time a user watches a recommended movie, you adjust your recommendation algorithms to better predict and
      meet future user preferences.
17 Your goal is to enhance the user's viewing experience by providing personalized and engaging movie suggestions.
18
19 A good strategy to optimize for reward in these situations requires balancing exploration
20 and exploitation. You need to explore to try out different movies and find those
21 with high rewards, but you also have to exploit the information that you have to
22 accumulate rewards.
23
24 So far you have interacted 2 times with the most recent following choices and rewards:
25 Context: a person who is a 18-year-old man with an occupation of college/grad student and live in Pulaski county,
      AR. The user has some numerical values that represent their true implicit preference or taste for all
      movies: [-0.011492758058011532, 0.027099572122097015, -0.020118921995162964, -0.002230832353234291,
      -0.003236030228435993].
26 Side Information for decision making:
27 {"American Beauty (1999)": {"exploration value": 0.018}, {"exploitation value":0.000}}
28 {"Star Wars: Episode IV - A New Hope (1977)": {"exploration value": 0.018}, {"exploitation value":0.000}}
29 {"Star Wars: Episode V - The Empire Strikes Back (1980)": {"exploration value": 0.018}, {"exploitation
      value":0.000}}
30 {"Star Wars: Episode VI - Return of the Jedi (1983)": {"exploration value": 0.018}, {"exploitation value":0.000}}
31 {"Jurassic Park (1993)": {"exploration value": 0.018}, {"exploitation value":0.000}}
32 {"Saving Private Ryan (1998)": {"exploration value": 0.018}, {"exploitation value":0.000}}
33 {"Terminator 2: Judgment Day (1991)": {"exploration value": 0.018}, {"exploitation value":0.000}}
34 {"The Matrix (1999)": {"exploration value": 0.018}, {"exploitation value":0.000}}
35 {"Back to the Future (1985)": {"exploration value": 0.018}, {"exploitation value":0.000}}
36 {"The Silence of the Lambs (1991)": {"exploration value": 0.018}, {"exploitation value":0.000}}
37 Action: The Silence of the Lambs (1991)
38 Reward: 4.121133 out of 5
39
40 Context: a person who is a 25-year-old man with an occupation of sales/marketing and live in Solano county, CA.
      The user has some numerical values that represent their true implicit preference or taste for all movies:
      [-0.00312434253282845, 0.0017211971571668983, 0.0015880014980211854, 0.012064018286764622,
      0.0090617602691105434].
41 Side Information for decision making:
42 {"American Beauty (1999)": {"exploration value": 0.008}, {"exploitation value":0.000}}
43 {"Star Wars: Episode IV - A New Hope (1977)": {"exploration value": 0.008}, {"exploitation value":0.000}}
44 {"Star Wars: Episode V - The Empire Strikes Back (1980)": {"exploration value": 0.008}, {"exploitation
      value":0.000}}
45 {"Star Wars: Episode VI - Return of the Jedi (1983)": {"exploration value": 0.008}, {"exploitation value":0.000}}
46 {"Jurassic Park (1993)": {"exploration value": 0.008}, {"exploitation value":0.000}}
47 {"Saving Private Ryan (1998)": {"exploration value": 0.008}, {"exploitation value":0.000}}
48 {"Terminator 2: Judgment Day (1991)": {"exploration value": 0.008}, {"exploitation value":0.000}}
49 {"The Matrix (1999)": {"exploration value": 0.008}, {"exploitation value":0.000}}
50 {"Back to the Future (1985)": {"exploration value": 0.008}, {"exploitation value":0.000}}
51 {"The Silence of the Lambs (1991)": {"exploration value": 0.008}, {"exploitation value":-0.000}}
52 Action: American Beauty (1999)
53 Reward: 0 out of 5
54
```

Figure A14: Contextual Bandit: Movie Recommendation for 10 movies, with Algorithm-Guided Support (Part 1)

```
 1 Context: a person who is a 56-year-old man with an occupation of sales/marketing and live in Jefferson county,
      KY. The user has some numerical values that represent their true implicit preference or taste for all
      movies: [-0.009686884470283985, 0.028794225305318832, -0.011435767635703087, 0.006439171731472015,
      -0.010343835689127445].
 2 Side Information for decision making:
 3 {"American Beauty (1999)": {"exploration value": 0.017}, {"exploitation value":-0.000}}
 4 {"Star Wars: Episode IV - A New Hope (1977)": {"exploration value": 0.017}, {"exploitation value":0.000}}
 5 {"Star Wars: Episode V - The Empire Strikes Back (1980)": {"exploration value": 0.017}, {"exploitation
      value":0.000}}
 6 {"Star Wars: Episode VI - Return of the Jedi (1983)": {"exploration value": 0.017}, {"exploitation value":0.000}}
 7 {"Jurassic Park (1993)": {"exploration value": 0.017}, {"exploitation value":0.000}}
 8 {"Saving Private Ryan (1998)": {"exploration value": 0.017}, {"exploitation value":0.000}}
 9 {"Terminator 2: Judgment Day (1991)": {"exploration value": 0.017}, {"exploitation value":0.000}}
10 {"The Matrix (1999)": {"exploration value": 0.017}, {"exploitation value":0.000}}
11 {"Back to the Future (1985)": {"exploration value": 0.017}, {"exploitation value":0.000}}
12 {"The Silence of the Lambs (1991)": {"exploration value": 0.017}, {"exploitation value":0.005}}
13 Action: The Silence of the Lambs (1991)
14 Reward: 3.9708314 out of 5
15
16 Context: a person who is a 25-year-old man with an occupation of executive/managerial and live in Washington
      county, DC. The user has some numerical values that represent their true implicit preference or taste for
      all movies: [-0.010095382109284401, 0.010144174098968506, -0.01811344549059868, -0.009553882293403149,
      -0.012143188156187534].
17 Side Information for decision making:
18 {"American Beauty (1999)": {"exploration value": 0.014}, {"exploitation value":0.000}}
19 {"Star Wars: Episode IV - A New Hope (1977)": {"exploration value": 0.014}, {"exploitation value":0.000}}
20 {"Star Wars: Episode V - The Empire Strikes Back (1980)": {"exploration value": 0.014}, {"exploitation
      value":0.000}}
21 {"Star Wars: Episode VI - Return of the Jedi (1983)": {"exploration value": 0.014}, {"exploitation value":0.000}}
22 {"Jurassic Park (1993)": {"exploration value": 0.014}, {"exploitation value":0.000}}
23 {"Saving Private Ryan (1998)": {"exploration value": 0.014}, {"exploitation value":0.000}}
24 {"Terminator 2: Judgment Day (1991)": {"exploration value": 0.014}, {"exploitation value":0.000}}
25 {"The Matrix (1999)": {"exploration value": 0.014}, {"exploitation value":0.000}}
26 {"Back to the Future (1985)": {"exploration value": 0.014}, {"exploitation value":0.000}}
27 {"The Silence of the Lambs (1991)": {"exploration value": 0.014}, {"exploitation value":0.006}}
28 Action: The Silence of the Lambs (1991)
29 Reward: 1.0985798 out of 5
30
31
32 You have a new user: PLEASE RESPOND ONLY WITH A CHOICE of MOVIES LISTED ABOVE AND NO TEXT EXPLANATION.
33
34 Context: This person is a 35-year-old man, working as a lawyer and live in Camden county, NJ. The user has some
      numerical values that represent their true implicit preference or taste for all movies:
      [-0.009149148128926754, -0.00417252816259861, 0.011747784912586212, -0.012008273974061012,
      -0.006486567202955484].
35 Side Information for decision making:
36 {"American Beauty (1999)": {"exploration value": 0.010}, {"exploitation value":0.000}}
37 {"Star Wars: Episode IV - A New Hope (1977)": {"exploration value": 0.010}, {"exploitation value":0.000}}
38 {"Star Wars: Episode V - The Empire Strikes Back (1980)": {"exploration value": 0.010}, {"exploitation
      value":0.000}}
39 {"Star Wars: Episode VI - Return of the Jedi (1983)": {"exploration value": 0.010}, {"exploitation value":0.000}}
40 {"Jurassic Park (1993)": {"exploration value": 0.010}, {"exploitation value":0.000}}
41 {"Saving Private Ryan (1998)": {"exploration value": 0.010}, {"exploitation value":0.000}}
42 {"Terminator 2: Judgment Day (1991)": {"exploration value": 0.010}, {"exploitation value":0.000}}
43 {"The Matrix (1999)": {"exploration value": 0.010}, {"exploitation value":0.000}}
44 {"Back to the Future (1985)": {"exploration value": 0.010}, {"exploitation value":0.000}}
45 {"The Silence of the Lambs (1991)": {"exploration value": 0.010}, {"exploitation value":-0.001}}
46 Action:
47
```

Figure A15: Contextual Bandit: Movie Recommendation for 10 movies, with Algorithm-Guided Support (Part 2)

```
1    You are a video recommendation system powered by a bandit algorithm for an online streaming platform.
2    There are 5 videos available in your library, titled [A, B, AI, BS, E].
3    When a user logs into the platform, you select a video to recommend based on their viewing history and
      preferences.
4    You aim to engage the user by recommending videos that they are likely to watch.
5    Each time a user watches a recommended video, you update your recommendation model to refine future
      suggestions,
6    enhancing user satisfaction and platform engagement.
7
8    A good strategy to optimize for reward in these situations requires balancing exploration
9    and exploitation. You need to explore to try out all of the videos and find those
10   with high rewards, but you also have to exploit the information that you have to
11   accumulate rewards.
12
13   Here are some examples of optimal actions under different scenarios. Use them as hints to help you come up
      with better actions.
14   =======================
15   A video, reward 1
16   B video, reward 1
17   AI video, reward 1
18   BS video, reward 0
19   E video, reward 0
20   A video, reward 0
21
22   Which video will you choose next? PLEASE RESPOND ONLY WITH A, B, C, D, E AND NO TEXT EXPLANATION.
23   B
24   =======================
25   A video, reward 1
26   B video, reward 1
27   AI video, reward 1
28   BS video, reward 0
29   E video, reward 0
30   A video, reward 0
31   B video, reward 0
32   AI video, reward 1
33   AI video, reward 0
34
35   Which video will you choose next? PLEASE RESPOND ONLY WITH A, B, C, D, E AND NO TEXT EXPLANATION.
36   AI
37   =======================
38
39   So far you have played 6 times with the following choices and rewards:
40   A video, reward 1
41   B video, reward 1
42   AI video, reward 1
43   BS video, reward 0
44   E video, reward 0
45   A video, reward 0
46
47   Which video will you choose next? PLEASE RESPOND ONLY WITH A, B, AI, BS, E AND NO TEXT EXPLANATION.
48
```

Figure A16: Multi-Arm Bandit: Bernoulli, Video Action Description, $K = 5$, Raw History, with In-context Few-shot Demonstrations from Bernoulli, Video Action Description, $K = 5$, Raw History.

```
1    You are a video recommendation system powered by a bandit algorithm for an online streaming platform.
2    There are 5 videos available in your library, titled [A, B, AI, BS, E].
3    When a user logs into the platform, you select a video to recommend based on their viewing history and
      preferences.
4    You aim to engage the user by recommending videos that they are likely to watch.
5    Each time a user watches a recommended video, you update your recommendation model to refine future
      suggestions,
6    enhancing user satisfaction and platform engagement.
7
8    A good strategy to optimize for reward in these situations requires balancing exploration
9    and exploitation. You need to explore to try out all of the videos and find those
10   with high rewards, but you also have to exploit the information that you have to
11   accumulate rewards.
12
13   Here are some examples of optimal actions under different scenarios. Use them as hints to help you come up
      with better actions.
14   ========================
15   Midnight Mirage Trousers item, reward 1
16   Titanic Tempest Tunic item, reward 0
17   Infinite Impeccable Jacket item, reward 1
18   Supreme Spectrum Slippers item, reward 0
19   Bejeweled Bloom Blazer item, reward 0
20   Midnight Mirage Trousers item, reward 0
21
22   Which video will you choose next? PLEASE RESPOND ONLY WITH A, B, C, D, E AND NO TEXT EXPLANATION.
23   Infinite Impeccable Jacket
24   ========================
25   Midnight Mirage Trousers item, reward 1
26   Titanic Tempest Tunic item, reward 0
27   Infinite Impeccable Jacket item, reward 1
28   Supreme Spectrum Slippers item, reward 0
29   Bejeweled Bloom Blazer item, reward 0
30   Midnight Mirage Trousers item, reward 0
31   Infinite Impeccable Jacket item, reward 0
32   Midnight Mirage Trousers item, reward 0
33   Infinite Impeccable Jacket item, reward 0
34
35   Which video will you choose next? PLEASE RESPOND ONLY WITH A, B, C, D, E AND NO TEXT EXPLANATION.
36   Titanic Tempest Tunic
37   ========================
38
39   So far you have played 6 times with the following choices and rewards:
40   A video, reward 1
41   B video, reward 1
42   AI video, reward 1
43   BS video, reward 0
44   E video, reward 0
45   A video, reward 0
46
47   Which video will you choose next? PLEASE RESPOND ONLY WITH A, B, AI, BS, E AND NO TEXT EXPLANATION.
48
```

Figure A17: Multi-Arm Bandit: Bernoulli, Video Action Description, $K = 5$, Raw History, with Few-shot Demonstrations from Bernoulli, Clothes Action Description, $K = 5$, Raw History

