# OpenReview forum: "EVOLvE: Evaluating and Optimizing LLMs For In-Context Exploration"
_ICML.cc/2025/Conference — ICML 2025 poster_

### Official Review · Reviewer_edRQ · 2025-02-27

**Overall Recommendation:** 4

**Summary:**

This paper studies the problem of in-context exploration, where an LLM interacts with a bandit environment and decides its next action based on the given context. The authors propose a framework called BanditBench, which includes both multi-armed bandit and contextual bandit instances, and suggest two methods to improve LLM's exploratory behavior: inference-time algorithmic guided support and algorithmic distillation. Empirical evaluation shows that few-shot learning boosts Flash's performance but hurts Pro's, while fine-tuning significantly improves performance across all models.

**Claims And Evidence:**

Evaluating and enhancing the exploration capabilities of LLMs is a novel, interesting, and important research problem.

The claims made in the submission are supported by clear and convincing evidence. The empirical evaluations demonstrate the effectiveness of the proposed methods, and the theoretical analyses provide a solid foundation for the significance and value of the study, which is important and easy-to-read.

**Essential References Not Discussed:**

Some LLMs for exploration, although not under bandit tasks, should be discussed to explain the differences in exploration from a contextual perspective. This would provide a more comprehensive understanding of the context and exploration differences.

[1] Qu, Yun, et al. "Choices are more important than efforts: Llm enables efficient multi-agent exploration." arXiv preprint arXiv:2410.02511 (2024).

[2 ]Bai C, Zhang Y, Qiu S, et al. Online Preference Alignment for Language Models via Count-based Exploration[J]. arXiv preprint arXiv:2501.12735, 2025.

**Experimental Designs Or Analyses:**

The experimental designs and analyses are sound and valid. The empirical evaluation is thorough, and the comparison of different models via pairwise win rate provides a clear understanding of the performance improvements achieved by the proposed methods.

**Methods And Evaluation Criteria:**

The proposed methods and evaluation criteria make sense for the problem at hand. The introduction of the BanditBench framework and the use of both multi-armed bandit and contextual bandit instances are appropriate for evaluating the exploration capabilities of LLMs.

**Other Comments Or Suggestions:**

N/A

**Other Strengths And Weaknesses:**

Pros:
1. The structure of the paper is clear and well-organized, making it easy to read.
2. The introduction of the benchmark makes a significant contribution to research in this field.

**Questions For Authors:**

1. Do the methods proposed in this paper still apply to in-context exploration on a larger scale?
2. Does in-context exploration have broader applicability, such as in the RLHF process?

**Relation To Broader Scientific Literature:**

(Krishnamurthy, 2024) has also studied in-context exploration. This paper builds upon that work by optimizing it, expanding the range of tasks, and emphasizing the importance of in-context exploration. However, there seem to be no surprising or novel findings, which might affect the paper's innovation aspect. Nonetheless, this does not negate the value of the paper, as its contributions in terms of usability are substantial.

**Theoretical Claims:**

The theoretical claims regarding the sufficiency of contextualization and regret are well-supported by the analyses provided in the paper. There are no apparent issues with the correctness of the proofs.

---

> ### Author Rebuttal · Authors · 2025-04-01
>
> We appreciate your thoughtful review and thank you for highlighting our contributions.
>
> > Some LLMs for exploration, although not under bandit tasks, should be discussed to explain the differences in exploration from a contextual perspective.
>
> Here we provide some comments on these two recent works.
>
> > [1] Qu, Yun, et al. "Choices are more important than efforts: Llm enables efficient multi-agent exploration." arXiv preprint arXiv:2410.02511 (2024).
>
> > [2] Bai C, Zhang Y, Qiu S, et al. Online Preference Alignment for Language Models via Count-based Exploration. arXiv preprint arXiv:2501.12735, 2025.
>
> Both works are about designing exploration bonus. First paper is using LLM to design such a bonus to train Deep RL models (in multi-agent setting). Second paper uses pseudo-count as exploration bonus to train LLM models. Our goal (algorithm distillation) is entirely different – we are trying to see if through prompting and supervised fine-tuning, we can distill the exploration behaviors into the LLM weights. This is an entirely new task / capability for the LLMs, similar to how to train LLM to generate content that matches human preference – we are training LLMs to explore optimally.
>
> > Do the methods proposed in this paper still apply to in-context exploration on a larger scale?
>
> We would argue that MovieLens (Movie Recommendation) is a realistic task on a much larger scale (the dataset we used has 1M real user ratings, 6000 users, over 4000 movies). We hope future work will explore other settings on an even larger scale as well.
>
> > Does in-context exploration have broader applicability, such as in the RLHF process?
>
> The most natural application/need for in-context exploration is actually in the LLM agent space. Many current agent applications require a planner [1] [2]. The planner generates a multi-step plan for a given task / user input, and then it gets executed. If this plan fails, a new plan will be proposed. In order to achieve a high task success rate, a planner (usually an LLM) needs to be able to explore plans efficiently (conditioned on past failed plans) in order to find the best. This is very similar to the bandit setup we have explored (push a button, see reward, try again). Such try-observe-retry loop is crucial to generalize out-of-distribution to unseen tasks or unseen scenarios.
>
> [1] Wang, Guanzhi, Yuqi Xie, Yunfan Jiang, Ajay Mandlekar, Chaowei Xiao, Yuke Zhu, Linxi Fan, and Anima Anandkumar. "Voyager: An open-ended embodied agent with large language models." arXiv preprint arXiv:2305.16291 (2023).
>
> [2] OWL: Optimized Workforce Learning for General Multi-Agent Assistance in Real-World Task Automation

---

> > ### Comment · Reviewer_edRQ · 2025-04-06
> >
> > Confirmed, and the score has been increased.

---

### Official Review · Reviewer_za2N · 2025-03-11

**Overall Recommendation:** 4

**Summary:**

This study investigates how the LLMs explore in context. The paper used Gemma/Gemini family of models of varying sizes, and bandit tasks (i.e., multi-armed bandit task and contextual bandit task) to evaluate the model's exploration behavior by in-context learning. Results show that all LLMs deviate from the optimal exploration algorithm. The authors used multiple approaches to try improving LLM's exploration behaviors, including using summarized history in the context, inserting optimal behavior as examples, or even fine-tuning models on the optimal traces on a similar but different scenario from the test. Those approaches showed various performance in improving LLM's exploration behavior. Overall, this paper highlights the fallback of LLMs' exploration capacity and proposed various approaches to mitigating such suboptimality.

**Claims And Evidence:**

The paper claims impact of task difficulty, contextualization and the generalization. Though the authors list the results and visualizes them, the comparison between metrics has not been tested by statistical approaches, which should be supplemented.

**Essential References Not Discussed:**

This paper lacks background about exploration behavior. Since the paper's key contribution is to propose a benchmark of LLM bandit exploration, the discussion about the exploration behavior (not only the final performance) is also important. For example, there is a number of human exploration behavior literatures which can be referred:

Gershman, S. J. Deconstructing the human algorithms for exploration. Cognition, 173:34–42, 2018.

Wilson, R. C., Geana, A., White, J. M., Ludvig, E. A., and Cohen, J. D. Humans use directed and random exploration to solve the explore–exploit dilemma. Journal of experimental psychology: General, 143(6):2074, 2014.

Daw, N. D., O’doherty, J. P., Dayan, P., Seymour, B., and Dolan, R. J. Cortical substrates for exploratory decisions in humans. Nature, 441(7095):876–879, 2006.

Citing these papers is not necessary. They are just providing some ways of capturing LLMs exploration behavior in a more subtle dynamic.

**Experimental Designs Or Analyses:**

The authors choose two types of bandit tasks and test them on the Gemma/Gemini model family. Several optimization approaches, including summarized history, few-shot learning, and oracle behavior fine-tuning, are set up for comparison. These models and tasks are relatively robust and comprehensive.

However, it is unknown to readers how the model is configured to generate responses (e.g., temperature, top K or top P parameters). Notably, these parameters, especially temperatures themselves, can have an impact on the explorative behaviors. How they are set up in a reasonable manner and how they would be controlled should be clearly identified in the paper.

**Methods And Evaluation Criteria:**

The paper used cumulative regret to evaluate the model's behavior. The key criterion is if the cumulative regret derivatives converge to 0 as the task goes on, the model is optimal. This makes sense since it can measure whether the model has finally explored the optimal bandit. The more the regret is increasing, the more suboptimal the model is.

However, there might be a possibility that the model rarely explores that happens to be at the optimal bandit, which may still yield optimal performance. Therefore, the dynamics of exploration behavior should also be considered in the evaluation. For example, how often the model switches their choices (something like win-stay-lose-switch), and when they switch, they tend to randomly choose, or choose less explored ones (random exploration vs. directed exploration). This would be more informative to not only reveal the performance but also the model's behaviors, especially help to better understand how each optimization approach works (since they may work from completely different aspects!)

**Other Comments Or Suggestions:**

The authors mentioned inference time guided (which is actually few-shot prompting) in the optimization. I was wondering if a reasoning model, or Chain-of-Thought prompting, could make exploration better. In a recent relevant paper, they found that stronger reasoning models can explore much better than traditional LLMs (Pan, Xie, Wilson, 2025). This discovery could potentially generalize to bandit tasks as well. The authors may try some reasoning models like o3-mini or deepseek-r1, QwQ32B and Gemini thinking. These reasoning models may bring unexpected good performance.

Another suggestion is to include the impact of temperatures on the explorative behavior. Rather than simply maximizing the token probability, adding moderate noises would push the model out for better rewards.

**Other Strengths And Weaknesses:**

Strengths: the paper has neat and clear writing as well as visualization. The optimization approach is diverse and can represent different mainstream pipelines.

Weakness:
1. As described above, one important fallback is that the exploration behavior is roughly described by regret analysis, which may overlook the dynamics in the exploit-exploration behaviors.

2. Due to 1, we may not know how different optimization approaches may improve the model's exploration behaviors. Some interpretability work can be referenced here:

Demircan, C., Saanum, T., Jagadish, A. K., Binz, M., and Schulz, E. Sparse autoencoders reveal temporal difference learning in large language models. arXiv preprint arXiv:2410.01280, 2024.

**Questions For Authors:**

1. I am curious why finetuning models with oracle behaviors could improve the model exploration behavior, even though the test scenario is never seen. What is a learned pattern? For example, is it just remembering to switch bandits in a fixed manner or adjust their behavior in an online manner. Since the mab's property is fixed over time, randomly sampling every choice will most likely give a shot on the optimal bandit. But if fine-tuning only implements a pattern recognition, it may fail in a changing bandit task (the property of each bandit is changing over time), which definitely requires more frequent exploration rather than remembering the sequences.

**Relation To Broader Scientific Literature:**

Large Language Models have been extensively evaluated on a variety of benchmarks, showcasing their pros and cons as a potential intelligent system. Exploration is one topic that is surprisingly overlooked but important to evaluate LLM's capacity. This topic is also a bridge between LLM and reinforcement learning, which should be important when combining these two to develop stronger model and even agents.

**Theoretical Claims:**

The main theoretical claims in the paper are about the definition of optimal behavior in MAB, which they used Upper Bound Confidence (UBC), a common algorithm for exploration in RL. The main argument for optimal behavior is to have cumulative regret derivatives converge to 0. Both algorithms and concepts are commonly used in the RL field and I don't see any problems.

---

> ### Author Rebuttal · Authors · 2025-04-01
>
> Thank you for the thoughtful reviews.
>
> > the comparison between metrics has not been tested by statistical approaches, which should be supplemented.
>
> The win-rate we calculated is actually after the Student’s t-test. We report this in Section 6.1 Metrics (Page 6). For each model on a given task, since we run 30 trials, we conduct the Student’s t-test on the cumulative reward over T-steps between two models  with p < 0.05 (Line 279).
>
> > it is unknown to readers how the model is configured to generate responses (e.g., temperature, top K or top P parameters)
>
> Thank you for bringing this up!
>
> We report this in Appendix Sec A.13: we use standard API calls and set the sampling temperature to 1.0 (range=[0.0, 2.0]). So give more context, the default API uses Top-P=0.95 sampling, and Top-K=40. [API Config file](https://github.com/google-gemini/deprecated-generative-ai-python/blob/main/google/generativeai/types/generation_types.py#L93) and [Doc](https://ai.google.dev/gemini-api/docs/text-generation). We will refer to this section in the main paper and discuss more about the exact set up.
>
> > This paper lacks background about exploration behavior.
>
> The list of papers you suggested are very helpful. We will include them in the paper.
>
> > However, there might be a possibility that the model rarely explores that happens to be at the optimal bandit, which may still yield optimal performance. Therefore, the dynamics of exploration behavior should also be considered in the evaluation.
>
> We appreciate the reviewer’s point that a model could, in theory, achieve high performance by consistently choosing the optimal arm—even if it rarely explores—by chance. To address this, we include an analysis of the model's exploration behavior using a metric called **OptFrac**, which measures how often the optimal arms are selected. As shown in Table 3, UCB steadily increases its OptFrac over time (32.7% → 49.4% → 58.7% → 62.6% → 65.0% over 1000 steps), indicating a growing focus on the optimal arm. In contrast, Gemini-1.5 Flash remains largely flat (9.3% → 10.1% → 10.4% → 10.6% → 10.7%), suggesting that it does not significantly shift its behavior toward the optimal arm. This supports our claim that the model does not accidentally achieve optimal performance by randomly selecting the best arm without meaningful exploration.
>
> > For example, how often the model switches their choices (something like win-stay-lose-switch), and when they switch, they tend to randomly choose, or choose less explored ones (random exploration vs. directed exploration).
>
> We appreciate the reviewer’s suggestion to analyze exploration dynamics in more detail, such as whether the model engages in random or directed exploration. In our analysis, we include a metric called **MinFrac**, which measures the fraction of pulls allocated to the least-selected arm. This captures the extent to which the model explores less-visited options—corresponding to what the reviewer refers to as “directed exploration.” Ideally, this value should be high early on (indicating strong directed exploration), and then decrease as the model gains experience and focuses on better-performing arms.
>
> As shown in Table 3, UCB exhibits this expected trend, with MinFrac values decreasing over time: 82.3% → 48.6% → 27.8% → 19.6% → 15.3%. In contrast, Gemini-1.5 Flash starts with a much lower MinFrac and declines rapidly (11.3% → 4.5% → 2.3% → 1.5% → 1.1%), suggesting it lacks meaningful directed exploration from the outset. We discussed this in Appendix Sec A.11.
> We know that we have only scratched the surface of understanding the dynamics of exploration. We hope our benchmark and work will inspire more investigations in the future.
>
> > Some interpretability work can be referenced here.
>
> Thank you—we'll add this to future work. Our focus is on evaluating and optimizing current model capabilities using standard prompting and fine-tuning techniques. We believe our benchmark can lay the groundwork for future interpretability and cognitive science research into LLM decision-making.
>
> > I was wondering if a reasoning model, or Chain-of-Thought prompting, could make exploration better.
>
> Thank you for the suggestion! Doing a full evaluation of thinking models is in our plan. We weren’t able to include results for this submission because most thinking models were released on the week of or after ICML deadline. We did a small scale investigation with o3-mini on Gaussian Multi-Arm bandit with 20 arms – thinking models are demonstrating stronger exploration capabilities. However, we need more time/resources to do a full-scale investigation.
>
> We appreciate the reference to (Pan, Xie, Wilson, 2025) – will include it in related work.
>
> There are still a lot of open questions – we share your excitement in exploring them. Hope we have addressed your questions and we are happy to answer more if they come up!

---

> > ### Comment · Reviewer_za2N · 2025-04-03
> >
> > Thanks to the authors for the rebuttal. The rebuttal mostly addresses my concerns and I will update my score to 4.

---

### Official Review · Reviewer_ZdZb · 2025-03-13

**Overall Recommendation:** 4

**Summary:**

This paper examines the ability of large language models (LLMs) to perform decision-making tasks, focusing on Multi-Armed Bandit (MAB) and Contextual Bandit (CB) problems. The authors introduce BanditBench, a benchmark suite for evaluating LLM decision-making capabilities in bandit environments. Additionally, the paper proposes two approaches to enhance LLM exploration: (1) inference-time algorithmic guided support and (2) algorithmic distillation through in-context demonstrations and fine-tuning using synthetic data generated from optimal algorithms. The empirical results reveal interesting behaviors of LLM agents in bandit tasks, offering valuable insights for future research.

**Claims And Evidence:**

The claims made in the paper are generally well-supported by empirical evidence. The introduction of BanditBench is a notable contribution, justified by the lack of standardized benchmarks in this area. The authors conduct thorough empirical evaluations and ablation studies, supporting their claim that BanditBench provides a structured framework for evaluating LLMs in decision-making under uncertainty. However, while the paper claims to propose novel methods for enhancing LLM decision-making, the Optimal Behavior Fine-Tuning approach closely resembles standard Behavioral Cloning (which is acknowledged), and the In-Context Few-Shot Demonstration technique is akin to in-context behavioral cloning. This similarity raises some concerns regarding the novelty of these contributions.

**Essential References Not Discussed:**

Not as far as I know of

**Experimental Designs Or Analyses:**

The experimental design appears sound, with thorough empirical evaluations conducted on the proposed benchmark. The authors perform ablation studies to analyze different aspects of their approach. However, as previously mentioned, the novelty of Optimal Behavior Fine-Tuning and In-Context Few-Shot Demonstration is a bit questionable.

**Methods And Evaluation Criteria:**

The proposed methods and evaluation criteria align well with the problem at hand. BanditBench provides a necessary and structured framework for assessing LLMs' decision-making and exploration capabilities. The empirical evaluation includes comprehensive ablation studies, strengthening the validity of the results. However, it would be useful for the authors to discuss the generalizability of BanditBench beyond bandit settings, particularly in more complex decision-making environments such as Markov Decision Processes (MDPs).

**Other Comments Or Suggestions:**

NA

**Other Strengths And Weaknesses:**

A particular strength is that the paper introduces a standardized benchmark, which in my opinion is crucial for enabling comparability across different studies in this domain.

The weakness of the paper is the lack of novelty in the proposed methods for LLM agents. But it is not even the main point of the paper.

**Questions For Authors:**

NA

**Relation To Broader Scientific Literature:**

The paper situates itself within the growing body of research on LLM agents in decision-making.

**Theoretical Claims:**

The paper does not focus on theoretical contributions, and no formal proofs are presented. Thus, no correctness checks on theoretical claims were required.

---

> ### Author Rebuttal · Authors · 2025-04-01
>
> Thank you for the review and your willingness to support our work!
>
> We appreciate the feedback and we are happy to answer questions if they come up!

---

### Official Review · Reviewer_FYEe · 2025-03-13

**Overall Recommendation:** 4

**Summary:**

The paper introduces BanditBench, a benchmark for in-context exploration using LLMs, and empirically investigates the ability of LLMs to explore using this benchmark. The paper also investigates ways to improve models ability to explore, using either in-context support from a bandit algorithm, or algorithm distillation (learning to imitate a bandit algorithm).

The paper shows that LLMs struggle with in-context exploration when the model is presented with the raw history of interactions, but that their algorithm-guided and algorithm-distillation methods significantly improve results. Finetuning works well, and can even make a small Gemini-1.5 Flash model outperform much larger models that have not been finetuned. Regret analyses show that larger models nearly get sub-linear regret when provided with algorithm guidance or a summarized history, while smaller models stay in the linear regime. Smaller models also achieve sublinear regret when finetuned. Algorithm guidance is particularly effective in the contextual setting. Nevertheless, significant gaps remain between classical (optimal) bandit algorithms and LLMs.

**Claims And Evidence:**

The claims in the paper are all well-supported. The authors do not aim to prove the superiority of their method but rather simply try to understand which factors affect in-context exploration ability, and in my opinion they did this in a rigorous way.

**Essential References Not Discussed:**

no

**Experimental Designs Or Analyses:**

The experiments are straightforward and well designed

**Methods And Evaluation Criteria:**

The benchmark introduced in this paper seems useful as a way to assess in-context exploration ability.

**Other Comments Or Suggestions:**

n/a

**Other Strengths And Weaknesses:**

The paper could do a better job discussing the long-term vision of this research. Since theoretically optimal bandit algorithms already exist, there seems to be no reason to use LLMs in cases where such algorithms can be applied. At the same time, it might be useful to have exploration ability in natively in the LLM, for unforeseen situations that may arise where bandits can't be directly applied. If that is the ultimate aim, then a big question is left open as to whether (finetuned) LLMs can in fact generalize their exploration ability from the synthetic setups studied in this paper to "real world" situations.

**Questions For Authors:**

It might be interesting to see how far one can get by training a randomly initialized small model with algorithm distillation. Is pre-training doing a lot of work?

**Relation To Broader Scientific Literature:**

The authors present the first Benchmark + analysis of in-context exploration of LLMs with bandits. Related works on RL algorithm distillation are cited and discussed in a balanced manner.

**Theoretical Claims:**

No substantial theoretical claims are made

---

> ### Author Rebuttal · Authors · 2025-04-01
>
> Thank you for the review! We appreciate your effort!
>
> > It might be interesting to see how far one can get by training a randomly initialized small model with algorithm distillation. Is pre-training doing a lot of work?
>
> We agree with your intuition. Pre-training indeed provides the right amount of bias that enabled us to fine-tune with a very small amount of data (50 trajectories for MAB with 300 steps to 1000 steps – 15000 to 50000 interactions/data points in total). Training from scratch would require a lot more interaction data and might not be able to generalize from one domain (i.e., "Video Watching") to another (i.e., "Clothes Shopping").
>
> It might be worth pointing out that most of the previous algorithm distillation (AD) work trained smaller Transformer models [1] [2] from scratch. We are one of the first works that fine-tuned a large language model instead and show AD indeed outperforms other prompting-based methods for in-context exploration.
>
> [1] Laskin, Michael, et al. "In-context reinforcement learning with algorithm distillation." arXiv preprint arXiv:2210.14215 (2022).
>
> [2] Lee, Jonathan, et al. "Supervised pretraining can learn in-context reinforcement learning." Advances in Neural Information Processing Systems 36 (2023): 43057-43083.

---

### Decision · Program_Chairs · 2025-05-01

**Decision:**

Accept (poster)

**Comment:**

The paper studies LLM's ability to perform exploration under a set of tasks about state-less bandit and contextual bandits. The paper also studies how to enable LLM to explore via explicit algorithm-guided support during inference and via distillation using in-context demonstrations and fine-tuning.

The reviewers are in general all positive and support for accpetance. Reviewers think that the introduction of the benchmark is very timely and is a good contribution, and claims made in the paper are in general well-supported by the empirical study.

There are a few things raised by reviewers which i think could be useful for the authors to discuss in the paper. First, discuss the long-term vision of this research -- theoretically optimal bandit algorithms does exist, and is training llm on the sythetic data generated by these algorithm on toy problems the ultimate way of making LLM learn to explore? Second, how can the work be extended to more challenging exploration settings such as the multi-step RL setting.